# VideoLoom: A Video Large Language Model for Joint Spatial-Temporal Understanding

**Jiapeng Shi** [1 2]  **Junke Wang** [1 2]  **Zuyao You** [1 2]  **Bo He** [3]  **Zuxuan Wu** [1 2]

## Abstract

Recent advancements in Video Large Language Models (Video LLMs) have demonstrated impressive results, yet existing approaches handle either temporal or spatial dimension in isolation, struggling in the analysis of complex events that require spatial-temporal integration. To bridge this gap, we propose VideoLoom, a unified Video LLM for joint spatial-temporal understanding. To facilitate the development of fine-grained spatial and temporal localization capabilities, we curate LoomData-8.7k, a character-centric video dataset with temporally grounded and spatially localized captions. With this, VideoLoom achieves state-of-the-art performance across a variety of spatial and temporal benchmarks. In addition, we introduce LoomBench, a benchmark consisting of temporal, spatial, and compositional video–question pairs, with a novel metric $\mathcal{J}\&\mathcal{F}_{bi\text{-}fore}$, enabling a comprehensive evaluation of Video LLMs from diverse aspects. Collectively, these contributions offer a universal and effective suite for joint spatial-temporal video understanding, setting a new standard in multimodal intelligence. Our code is available at https://github.com/JPShi12/VideoLoom.

## 1. Introduction

Recent years have witnessed the rapid development of Multimodal Large Language Models (MLLMs) (Hurst et al., 2024; Comanici et al., 2025; Bai et al., 2025; Chen et al., 2024), extending their scope from static image understanding (Liu et al., 2023; 2024a; Shao et al., 2024; Wang et al., 2023c; Meng et al., 2024) to dynamic video comprehension (Li et al., 2023; Maaz et al., 2024; Li et al., 2024b;

Wang et al., 2023a; Xie et al., 2026). Video Large Language Models (Video LLMs), which integrate spatial perception with temporal reasoning, have demonstrated strong generalization and competitive performance across a wide range of multimodal benchmarks. More recently, increasing efforts have been devoted to equipping Video LLMs with fine-grained understanding capabilities, such as temporal grounding (Ren et al., 2024; Huang et al., 2024a; Wang et al., 2024b), referring video segmentation (Yan et al., 2024; Yuan et al., 2025a; Deng et al., 2025; You & Wu, 2025), and object tracking (Zhu et al., 2023; Bai et al., 2024; Yang et al., 2024). Despite these achievements, most existing models still focus on either temporal or spatial dimension in isolation, limiting their ability to holistically interpret complex spatial-temporal events in real-world scenarios.

While joint spatial-temporal understanding represents a promising direction for Video LLMs, several critical challenges still remain. First and foremost, a fundamental limitation is the scarcity of high-quality datasets with fine-grained spatial-temporal annotations. Most existing datasets provide either temporal (e.g., event segments (Krishna et al., 2017; Zhou et al., 2018)) or spatial labels (e.g., object trajectories (Ding et al., 2023; Seo et al., 2020)), but rarely both. A straightforward practice is to jointly train on both types of datasets, but inconsistencies in annotation formats and data distributions often lead to unstable training and hinder the model from establishing coherent spatial–temporal associations. In addition, spatial and temporal video tasks inherently demand different input granularities, i.e., spatial tasks typically require higher resolutions to capture fine-grained details (Wang et al., 2025; 2023b; Yan et al., 2024; ?), while temporal tasks depend on denser frame sampling to model motion dynamics (Ren et al., 2024; Guo et al., 2025b). Under fixed computational budgets, it is difficult to balance both requirements, making joint spatial–temporal modeling within a single framework inherently challenging.

To address the above issues, we first introduce LoomData-8.7k, a novel dataset with consistent spatial and temporal annotations. LoomData-8.7k sources videos from ActivityNet (Caba Heilbron et al., 2015) and is annotated using an automatic pipeline. Specifically, we first segment each untrimmed video into multiple shots and then identify the

---

[1]Institute of Trustworthy Embodied AI, Fudan University, Shanghai, China [2]Shanghai Key Laboratory of Multimodal Embodied AI, Shanghai, China [3]University of Maryland, College Park, USA. Correspondence to: Zuxuan Wu <zxwu@fudan.edu.cn>.

*Proceedings of the 43rd International Conference on Machine Learning*, Seoul, South Korea. PMLR 306, 2026. Copyright 2026 by the author(s).

main characters in the initial shot. Based on this, we track trajectories and generate corresponding action descriptions for each character. This character-centric, shot-guided automatic annotation pipeline provides richer spatial references and complete temporal coverage, enabling detailed and coherent spatial–temporal understanding.

With this, we present VideoLoom, a simple yet effective Video LLM for joint spatial–temporal understanding. To accommodate both fine-grained capabilities within a single framework, we integrate an MLLM with SAM2 (Ravi et al., 2025) in an end-to-end manner, unifying the time-aware instruction modeling of MLLM and the superior spatial localization of SAM2. To balance temporal coverage and spatial precision, we introduce two types of visual tokens, i.e., fast tokens and slow tokens. The former are generated across the entire video span, providing global temporal context with a low token density per frame. The latter are extracted from keyframes assigned a higher token density. By interleaving SlowFast visual tokens with frame IDs, VideoLoom establishes a unified input sequence that facilitates coherent spatial-temporal comprehension, fostering mutual promotion across both dimensions.

To comprehensively evaluate the spatial–temporal understanding capability of Video LLMs, we also propose LoomBench, a benchmark comprising 130 videos and over 1,400 queries spanning temporal grounding and spatial segmentation, with a novel evaluation metric. Unlike existing datasets that assess these dimensions separately (Zhang et al., 2026; Ding et al., 2023) or remain at coarse granularity (Lei et al., 2020; Cheng et al., 2026), LoomBench consists of carefully designed questions that require models to perform grounding and segmentation simultaneously.

To summarize, this paper introduces the first suite to unlock joint spatial-temporal understanding capabilities across three essential aspects: dataset, model, and benchmark. Experiment results demonstrate that VideoLoom achieves new state-of-the-art across 10 visual benchmarks, including spatial benchmarks (e.g., 63.1 $\mathcal{J}\&\mathcal{F}$ on ReVOS (Yan et al., 2024)) and temporal ones (e.g., 48.3 R1@0.7 on Charades-STA (Gao et al., 2017), 63.3 HIT@1 on QVHighlights (Lei et al., 2021)). Comparisons with existing Video LLMs on LoomBench further validate the effectiveness of VideoLoom in unified spatial-temporal comprehension.

## 2. Related Work

### 2.1. Spatial-Temporal Video Datasets

Existing video datasets for spatial-temporal understanding can generally be categorized into two separate types: temporal-focused and spatial-focused. Temporal-focused datasets, e.g., for dense captioning (Zhou et al., 2018; Krishna et al., 2017) or temporal grounding (Gao et al.,

2017), provide descriptions aligned with timestamps but typically lack spatial annotations. In contrast, spatial-focused datasets focus on spatial localization through segmentation masks (Seo et al., 2020; Ding et al., 2023) or trajectory annotations (Fan et al., 2019; Huang et al., 2019), but do not include precise temporal locations of actions. Few datasets focus on atomic actions (Gu et al., 2018; Zhang et al., 2020) or grounded QA (Lei et al., 2020; Cheng et al., 2026), featuring spatiotemporal tubelets, yet constrained by coarse-grained annotations in the form of sparse bounding boxes. Additionally, current datasets rely on costly manual annotations with brief captions of objects or events, lacking detailed positional references and temporal coverage. Collectively, these factors hinder the training of Video LLMs with spatial-temporal comprehension. To bridge this gap, we introduce LoomData-8.7k, providing both fine-grained temporal annotations and mask-level spatial tracklets at scale, enabling more comprehensive spatial-temporal modeling.

### 2.2. Video Large Language Models

Recent advancements in Video LLMs reveal a clear trend from coarse-grained tasks such as captioning and retrieval (Shvetsova et al., 2024; Xu et al., 2025b) to fine-grained comprehension that captures precise object interactions and temporal dynamics (Wang et al., 2024a; Peng et al., 2025; Yuan et al., 2025b; Han et al., 2025). Within this landscape, Video LLMs designed for fine-grained video understanding can be broadly categorized into two directions: temporal-focused and spatial-focused models. Temporal models (Ren et al., 2024; Guo et al., 2025b) are trained on timestamp-aware instruction data to develop temporal localization capabilities. Spatial models (Yuan et al., 2025a; Gong et al., 2025), on the other hand, focus on grounding visual regions in the format of trajectories. While both directions address critical aspects of video understanding, neither is sufficient in isolation. Some works (Cheng et al., 2024; He et al., 2026) begin to model the spatial-temporal clues in video simultaneously. SF-LLaVA (Xu et al., 2024; 2025a) and LITA (Huang et al., 2024b) introduce SlowFast tokens to model long-form videos efficiently, but only for holistic comprehension or specific tasks. LLaVA-ST (Li et al., 2025a) and SpaceVLLM (Wang et al., 2026a) focus on spatiotemporal grounding, yet remain confined to coarse-grained perception with bounding boxes. In this paper, we propose a unified Video LLM, VideoLoom, which for the first time accommodates both fine-grained temporal understanding and spatial perception within a single framework.

## 3. Method

This section introduces the VideoLoom suite, which advances joint spatial-temporal understanding from three key perspectives: 1) **LoomData-8.7k**, a video dataset with fine-grained spatial and temporal annotations. 2) **VideoLoom**, a

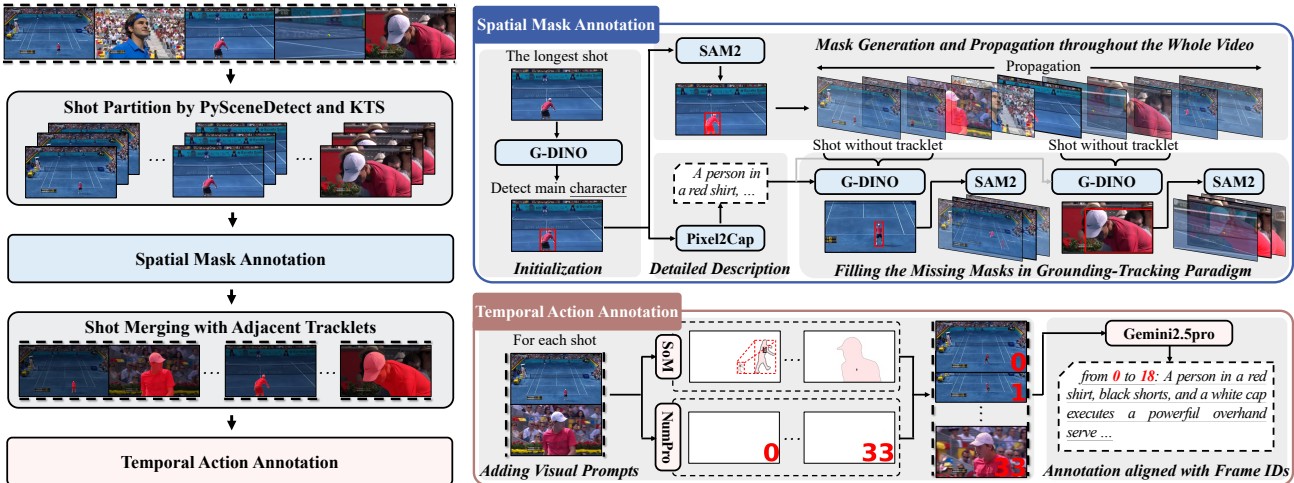

*Figure 1.* Illustration of the designed data annotation pipeline, comprising four stages: shot partition, spatial mask annotation, shot merging, and temporal action annotation. During spatial mask annotation, main characters and their complete tracklets are identified. In temporal action annotation, actions of characters are temporally grounded with visual prompts.

Video LLM that handles joint temporal understanding and spatial perception tasks within a single framework. and 3) **LoomBench**, a video benchmark developed to evaluate the joint spatial-temporal capability of Video LLMs.

### 3.1. LoomData-8.7k

We develop an automatic annotation pipeline that leverages multiple visual foundation models to detect and associate the temporal actions and spatial locations of main characters. As shown in Fig. 1, the pipeline comprises four main stages: (i) shot partition, (ii) spatial mask annotation, (iii) shot merging, and (iv) temporal action annotation.

**Shot Partition.** We first partition each video into several shots using PySceneDetect (Castellano, 2022) and KTS (Potapov et al., 2014). PySceneDetect identifies scene boundaries by detecting scene changes between adjacent frames, while KTS captures event transitions. We combine both by sequentially ordering the timestamps of all transition points to achieve accurate shot partition.

**Spatial Mask Annotation.** For each video, we use GroundingDINO (Liu et al., 2024b) to detect the "person" category in the center frame of the longest shot, and only keep the bounding box with the highest score as the main character. With this region box, a detailed description of its appearance (e.g., clothing and attributes) is generated by Pix2Cap (You et al., 2025). After this, we employ SAM2 (Ravi et al., 2025) to track the main character throughout the video, producing the initial mask tracklet. We then complete the cross-shot tracklet in a grounding-tracking paradigm. GroundingDINO is also applied to re-annotate this character in the center frame of shots without a tracklet, where initial tracking failed, based on the description. SAM2 then conducts mask tracking on these shots to fill the missing masks, yielding

a complete tracklet of the main character. Finally, we perform a manual verification step to refine redundantly tracked shots and remove incorrectly tracked videos. Please refer to Sec. C.1 for the workflow.

**Shot Merging.** As a temporal event may span multiple shots (e.g., different camera angles), we merge all adjacent shots annotated with tracklets to obtain temporally consistent annotations.

**Temporal Action Annotation.** With the merged shots and dense trajectories, we then generate detailed, timestamp-aligned action descriptions for main characters in each video. Specifically, we place unique numerical IDs on video frames sampled at 2 FPS in the manner of NumPro (Wu et al., 2025), and then employ Set-of-Marks (SoM) (Yang et al., 2023) to overlay an instance ID directly onto the segmentation masks of main characters. These sampled frames, along with both visual prompts, are fed into Gemini2.5pro (Comanici et al., 2025) to produce fine-grained action descriptions aligned with the frame IDs. Here, we employ similarity-based filtering for quality control. Details of this step and other automatic filtering rules used throughout the pipeline are provided in Sec. C.2.

We annotate the training set of ActivityNet (Caba Heilbron et al., 2015) with the above pipeline, resulting in 8,710 shots featuring both timestamp-aligned action descriptions and dense spatial masks. On average, each video has a duration of 102.2 seconds and includes 6.0 shots, while the temporal descriptions average 41.3 words. For additional statistics, please refer to Sec. C.3.

### 3.2. VideoLoom

With the above dataset, we further propose VideoLoom, a unified Video LLM to unlock further joint spatial-temporal under-

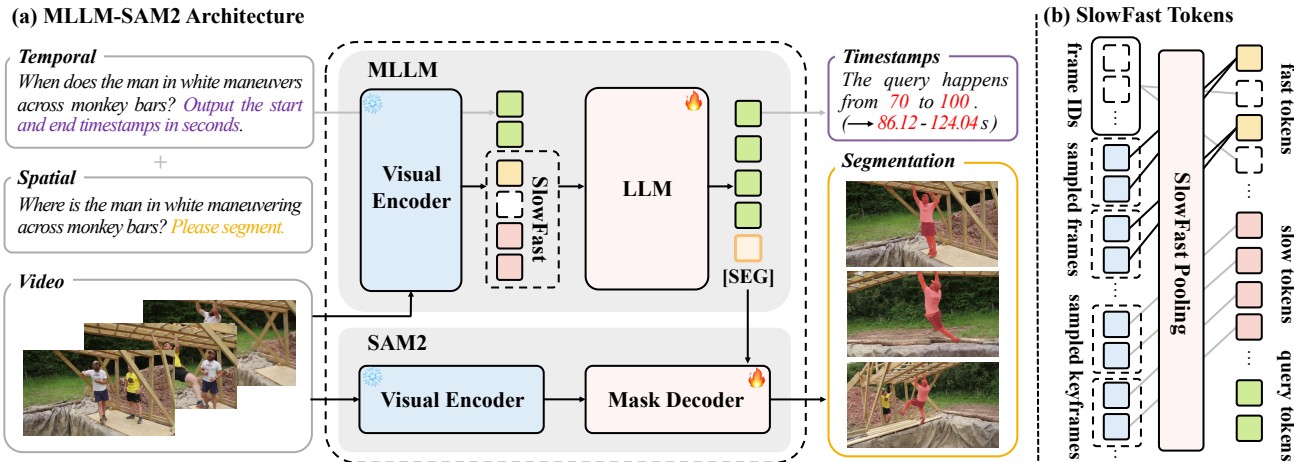

Figure 2. Overview of VideoLoom Architecture. Two key designs are: (a) MLLM-SAM2 Architecture, where MLLM and SAM2 are connected via a `[SEG]` token, unifying temporal understanding and spatial perception. (b) SlowFast Tokens, where input videos are encoded as SlowFast visual tokens to model spatial-temporal representations.

standing capabilities. Specifically, taking a language query $T$ and a video consisting of $N$ frames $V \in \mathbb{R}^{N \times H \times W \times 3}$ as input, where $H$ and $W$ denote the height and width of each frame respectively, VideoLoom aims to generate an answer text $O$ that contains the required timestamp information, or predict a trajectory in the format of segmentation masks $M \in \mathbb{R}^{N \times H \times W}$:

$$O, M = \text{VideoLoom}(T, V). \quad (1)$$

Below, we introduce the SlowFast visual tokens which capture spatial–temporal information at different granularities in Sec. 3.2.1, the MLLM-SAM2 architecture which integrates these tokens for unified spatial–temporal modeling in Sec. 3.2.2, and the loss functions in Sec. 3.2.3.

### 3.2.1. SLOWFAST VISUAL TOKENS

Temporal understanding typically requires processing a large number of frames (Ren et al., 2024; Huang et al., 2024a), whereas spatial perception demands higher-resolution inputs (Yuan et al., 2025a). To accommodate both, we introduce two types of visual tokens, i.e., fast tokens and slow tokens, which respectively encode dense frames with temporal bindings and sparse keyframes with rich spatial details.

Specifically, we sparsely sample $N_s$ keyframes and assign $C$ tokens for each frame to form $N_s \times C$ slow tokens. Meanwhile, we also densely sample $N_f$ frames across the entire video, paired with temporal indices. Both are fed to a visual encoder to obtain $N_s \times C$ slow tokens and $N_f \times \frac{C}{R^2}$ fast tokens, where the latter are processed via average pooling with a spatial downsampling ratio $R$.

### 3.2.2. MLLM-SAM2 ARCHITECTURE

**Overview.** We integrate InternVL3 (Zhu et al., 2025), a multimodal large language model (MLLM), with SAM2 (Ravi

et al., 2025), a video segmentation and tracking model, to support both spatial and temporal tasks within a unified framework. InternVL3 takes SlowFast visual tokens and text prompts as inputs, producing text responses, timestamps, and a `[SEG]` token embedding. SAM2 then utilizes this `[SEG]` token to generate corresponding segmentation masklets. The overall architecture is illustrated in Fig. 2.

**MLLM for temporal understanding tasks.** Our MLLM consists of a visual encoder, a visual projection layer, and an LLM. The sampled frames are input to the visual encoder and then mapped into visual tokens by the visual projection layer. Unlike previous work using absolute timestamps (Ren et al., 2024; Zeng et al., 2025) or special time tokens (Huang et al., 2024b; Guo et al., 2025b), we interleave unique frame IDs between visual tokens to indicate temporal order. The complete token sequence is used as input to the LLM, which models the spatial-temporal visual features and generates text token predictions according to text queries. Note that for timestamp-related queries, the LLM outputs corresponding frame IDs in text responses to indicate temporal locations.

**SAM2 for spatial understanding tasks.** Given the keyframes sampled for slow tokens, we input them to SAM2 at a higher resolution to predict spatial trajectories. A `[SEG]` token is used to connect MLLM with SAM2 mask decoder, providing the mask decoder with rich target information and prompting it to generate masks in the keyframes. We then propagate these masks to the entire video using a post-processing visual memory (Yuan et al., 2025a).

### 3.2.3. LOSS FUNCTIONS

VideoLoom is trained in an end-to-end manner with the following objective:

$$\mathcal{L} = \lambda_{\text{text}}\mathcal{L}_{\text{text}} + \lambda_{\text{mask}}\mathcal{L}_{\text{mask}}. \quad (2)$$

where $\mathcal{L}_{\text{text}}$ denotes the standard cross-entropy loss for

**When**

**Q:** When does the swimmer leap off the diving platform, executing a powerful and graceful dive into the pool?

**A:** *13.5s to 21.12s*

**Where**

**Q:** Where is the swimmer standing during 13.5s to 21.12s, extending her arms above her head and preparing for a dive?

**A:**

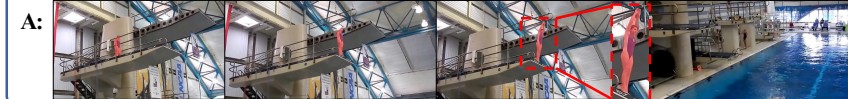

**Combined**

**Q:** Where is the swimmer when she emerges from the water, catching her breath after completing her dive, with her arms extended and legs bent?

**A:** *23.12s to 28.46s*

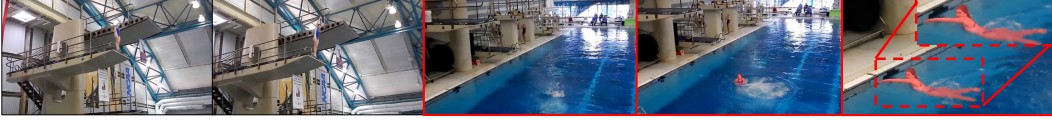

*Figure 3.* Visualization of the QA pairs in LoomBench. Three types of QA are shown: *When* targets the action timestamps given a query and the whole video, *Where* targets the person masklet given a query and a certain video segment, while *Combined* directly targets the tracklet segment corresponding to the query.

text generation, and $\mathcal{L}_{\text{mask}}$ indicates the segmentation loss combining per-pixel binary cross-entropy (BCE) loss and DICE loss. $\lambda_{\text{text}}$ and $\lambda_{\text{mask}}$ are balancing hyper-parameters.

### 3.3. LoomBench

We curate LoomBench, a new benchmark designed to jointly evaluate the spatial and temporal understanding capabilities of Video LLMs. Specifically, we apply the automatic annotation pipeline described in Sec. 3.1 to the validation set of ActivityNet (Caba Heilbron et al., 2015) to generate preliminary annotations. For each video shot, we prompt LLaMA3.1 (Grattafiori et al., 2024) to generate three types of questions based on the action descriptions of the main characters: *When*, *Where*, and *Combined*. Both the preliminary annotations and the generated questions are manually verified and refined to guarantee quality and consistency. As a result, LoomBench contains 130 videos, with an average of 4.2 temporal shots per video and an average shot length of 17.6 seconds. A visualization example is shown in Fig. 3.

***When/Where*** questions respectively target the action timestamps and the person masks of each segment, focusing on the evaluation of temporal understanding and spatial perception. Following existing benchmarks (Gao et al., 2017; Krishna et al., 2017), we adopt R1@0.5 and temporal IoU (tIoU) as evaluation metrics for *When* questions. For *Where* questions, we use the $\mathcal{J}\&\mathcal{F}$ metric (Seo et al., 2020; Ding et al., 2023), which averages region similarity ($\mathcal{J}$) and contour accuracy ($\mathcal{F}$). LoomBench contains 541 *When* and 487 *Where* questions.

***Combined*** questions such as "Where is the person when he/she is doing something?" extend beyond the scope of existing datasets and enable more comprehensive evaluation of unified spatial–temporal understanding. We annotate 456 *Combined* questions in LoomBench. The standard $\mathcal{J}\&\mathcal{F}$ metric computes the difference between predicted masklets and groundtruth across all video frames. However, for *Com-*

*bined* questions, the duration of the queried tracklet constitutes only a small fraction of the entire video (on average, 20.9%), making $\mathcal{J}\&\mathcal{F}$ dominated by background frames without mask annotations. These backgrounds can inflate $\mathcal{J}\&\mathcal{F}$ scores and undermine their reliability for evaluation. To address this issue, we propose Bidirectional Foreground $\mathcal{J}\&\mathcal{F}$, which computes $\mathcal{J}\&\mathcal{F}$ within the temporal intervals of both the predicted and groundtruth foreground masks, and then takes their harmonic mean:

$$\mathcal{J}\&\mathcal{F}_{bi\text{-}fore} = \frac{(\mathcal{J}_p + \mathcal{F}_p) \cdot (\mathcal{J}_g + \mathcal{F}_g)}{(\mathcal{J}_p + \mathcal{F}_p) + (\mathcal{J}_g + \mathcal{F}_g)}. \quad (3)$$

where $\mathcal{J}_p = \mathcal{J}_{\text{Loc}(P)}(P, G)$, $\mathcal{J}_g = \mathcal{J}_{\text{Loc}(G)}(P, G)$, $\mathcal{F}_p = \mathcal{F}_{\text{Loc}(P)}(P, G)$, and $\mathcal{F}_g = \mathcal{F}_{\text{Loc}(G)}(P, G)$. $P, G$ denote the predicted and groundtruth masks, and the function $\text{Loc}(\cdot)$ identifies the continuous temporal range of a masklet. Accordingly, $\mathcal{J}_p$ refers to the $\mathcal{J}$ score computed over the temporal segment of predicted masklet, and so on. For more analysis on $\mathcal{J}\&\mathcal{F}_{bi\text{-}fore}$, please refer to Sec. D.1.

## 4. Experiments

### 4.1. Experimental Setup

**Training data:** Our training data can be categorized into four types: 1) image question answering (QA), which includes LLaVA-665k (Liu et al., 2024a). 2) image segmentation data, comprising standard referring expression segmentation datasets (Kazemzadeh et al., 2014; Yu et al., 2016) and grounding conversation generation (GCG) data (Rasheed et al., 2024). 3) video segmentation data, including RefYTVOS (Seo et al., 2020), MeVIS (Ding et al., 2023), and ReVOS (Yan et al., 2024). 4) video temporal instruction data, consisting of Charades-STA (Gao et al., 2017), YouCook2 (Zhou et al., 2018), and QVHighlights (Lei et al., 2021). The proposed LoomData-8.7k is converted into both referring video object segmentation (VOS) and temporal grounding formats for joint training.

*Table 1.* Performance comparison on diverse temporal understanding benchmarks, [*] denotes models specifically designed for TVG.

| Method | Charades-STA | | YouCook2 | | | QVHighlights | |
|---|---|---|---|---|---|---|---|
| | R1@0.5 | R1@0.7 | S | C | F1 | mAP | HIT@1 |
| InternVL3-8B (Zhu et al., 2025) | 24.8 | 12.3 | 0.3 | 0.8 | 3.7 | 13.6 | 17.0 |
| TimeChat-7B (Ren et al., 2024) | 46.7 | 23.7 | 3.4 | 11.0 | 19.5 | 21.7 | 37.9 |
| VTG-LLM-7B (Guo et al., 2025a) | 57.2 | 33.4 | 3.6 | 13.4 | 20.6 | 24.1 | 41.3 |
| TRACE-7B (Guo et al., 2025b) | 61.7 | 41.4 | 6.7 | 35.5 | 31.8 | **31.8** | 51.5 |
| TimeSuite-7B (Zeng et al., 2025) | 67.1 | 43.0 | - | - | - | 27.0 | 55.3 |
| HawkEye-7B[*] (Wang et al., 2024b) | 58.3 | 28.8 | - | - | - | - | - |
| UniTime-7B[*] (Li et al., 2025b) | 75.3 | 56.9 | - | - | - | - | - |
| VideoLoom-8B | **70.0** | **48.3** | **7.3** | **41.5** | **33.6** | 27.5 | **63.3** |

*Table 2.* Performance comparison on ref-VOS.

| Method | MeVIS | RefYTVOS | ReVOS |
|---|---|---|---|
| | $\mathcal{J}\&\mathcal{F}$ | $\mathcal{J}\&\mathcal{F}$ | $\mathcal{J}\&\mathcal{F}$ |
| LISA-7B (Lai et al., 2024) | 37.2 | 53.9 | 40.9 |
| TrackGPT-7B (Zhu et al., 2023) | 40.1 | 56.4 | 43.6 |
| VISA-7B (Yan et al., 2024) | 43.5 | 61.5 | 46.9 |
| ViLLa-6B (Zheng et al., 2025) | 49.4 | 67.5 | 57.0 |
| GLUS-7B (Lin et al., 2025) | 51.3 | 67.3 | 54.9 |
| Sa2VA-8B (Yuan et al., 2025a) | 46.9 | 70.7 | 57.6 |
| VRS-HQ-7B (Gong et al., 2025) | 50.6 | 70.4 | 59.1 |
| VRS-HQ-13B (Gong et al., 2025) | 50.9 | 71.0 | 60.0 |
| VideoLoom-8B | **51.7** | **71.3** | **63.1** |

*Table 3.* Performance comparison on image segmentation benchmarks, including referring segmentation and GCG.

| Method | RC | RC+ | RCg | GCG | |
|---|---|---|---|---|---|
| | cIoU | cIoU | cIoU | AP50 | mIoU |
| VRS-HQ-7B (Gong et al., 2025) | 73.5 | 61.7 | 66.7 | - | - |
| LISA-7B (Lai et al., 2024) | 74.9 | 65.1 | 67.9 | - | - |
| OMG-LLaVA-7B (Zhang et al., 2024) | 78.0 | 69.1 | 72.9 | 29.9 | 65.5 |
| GLaMM-7B (Rasheed et al., 2024) | 79.5 | 72.6 | 74.2 | 30.8 | 66.3 |
| Sa2VA-8B (Yuan et al., 2025a) | 81.6 | 76.2 | 78.7 | 31.0 | - |
| VideoLoom-8B | **83.4** | **79.2** | **81.4** | **34.1** | **68.6** |

*Table 4.* Performance comparison on LoomBench.

| Method | When | | Where | Combined | |
|---|---|---|---|---|---|
| | R1 | tIoU | $\mathcal{J}\&\mathcal{F}$ | tIoU | $\mathcal{J}\&\mathcal{F}_{bi\text{-}fore}$ |
| TimeSuite-7B (Zeng et al., 2025) | 23.1 | 27.6 | - | - | - |
| Sa2VA-8B (Yuan et al., 2025a) | - | - | 86.1 | - | - |
| TimeSuite+Sa2VA | - | - | - | 25.4 | 33.7 |
| VideoLoom-8B | **37.9** | **39.7** | **87.2** | **41.6** | **49.1** |

**Implementation details:** We choose InternVL3 (Zhu et al., 2025) as our foundation MLLM and SAM2 (Ravi et al., 2025) as the segmentation module. A special token [SEG] is added for the mask generation following LISA (Lai et al., 2024). The input frames are resized to $448{\times}448$ and $1024{\times}1024$ for the MLLM and SAM2 visual encoders, respectively. The number of slow visual tokens $C$ per frame is set to 256, and the downsampling ratio $R$ is kept at 4, resulting in 16 fast tokens per frame. Both uniformly sampled, up to 128 frames are encoded as fast tokens, whereas only 5 keyframes for slow tokens. We use the XTuner (Contributors, 2023) codebase for training and evaluation, finetuning only the mask decoder and LLM module while keeping the visual encoder frozen. The LLM is adapted via LoRA (Hu et al., 2022), with a learning rate of $4 \times 10^{-5}$. The loss weights $\lambda_{\text{text}}$ and $\lambda_{\text{mask}}$ are both set to 1. We train VideoLoom for one epoch with a global batch size of 64. All experiments are facilitated on 8 NVIDIA H20 GPUs.

## 4.2. Main Results

**Comparison on Temporal Benchmarks.** We evaluate our model on a wide range of temporal tasks, including temporal

video grounding (TVG), dense video captioning (DVC), and video highlight detection (VHD), for a comprehensive assessment of its temporal understanding capabilities.

The comparison with existing Video LLMs is reported in Tab. 1. VideoLoom achieves state-of-the-art or competitive performance across TVG, DVC, and VHD, e.g., 48.3 R1@0.7 on Charades-STA, 7.3 SODA_c on YouCook2, and 63.3 HIT@1 on QVHighlights, surpassing both unified models, e.g., TimeSuite (Zeng et al., 2025), and task-specific models, e.g., HawkEye (Wang et al., 2024b). This highlights the strong temporal understanding capabilities of our method. Although VideoLoom lags behind UniTime (Li et al., 2025b) on Charades-STA, we attribute this to the much larger amount of grounding data used in their training and the complex inference procedure involving recursive localization.

**Comparison on Spatial Benchmarks.** For spatial understanding in videos, we evaluate our method on referring Video Object Segmentation (VOS) task on RefYTVOS (Seo et al., 2020), MeVIS (Ding et al., 2023), and ReVOS (Yan et al., 2024). $\mathcal{J}\&\mathcal{F}$ is chosen as the metric. The results in Tab. 2 show that VideoLoom even outperforms tracking-oriented Video LLMs on all these benchmarks, achieving 51.7 on MeVIS, 71.3 on RefYTVOS, and 63.1 on ReVOS in terms of $\mathcal{J}\&\mathcal{F}$. This superior performance showcases the effectiveness of our method for fine-grained spatial understanding.

Additionally, we also evaluate VideoLoom on image benchmarks, including RefCOCO (Kazemzadeh et al., 2014), RefCOCO+ (Kazemzadeh et al., 2014), and RefCOCOg (Yu et al., 2016) for referring segmentation, and Grand-f (Rasheed et al., 2024) for Grounded Conversation Generation (GCG). We adopt cIoU, AP50, and mIoU as the measurement metrics. The comparison results in Tab. 3 demonstrate that VideoLoom achieves the best results on all datasets, further demonstrating its strong spatial capabilities.

**Comparison on LoomBench.** Finally, we evaluate the joint spatial-temporal comprehension capability on the proposed LoomBench. For comparison, we design a strong baseline

*Table 5.* Ablation experiments on SlowFast visual tokens.

| Setting | Charades-STA | | | YouCook2 | | QVHighlights | | MeVIS | RefYTVOS | ReVOS | | |
|---|---|---|---|---|---|---|---|---|---|---|---|---|
| | R1@0.5 | R1@0.7 | mIoU | S | F1 | mAP | mIoU | $\mathcal{J}\&\mathcal{F}$ | $\mathcal{J}\&\mathcal{F}$ | $\mathcal{J}\&\mathcal{F}_{\text{Ref.}}$ | $\mathcal{J}\&\mathcal{F}_{\text{Rea.}}$ | $\mathcal{J}\&\mathcal{F}$ |
| Spatial (Slow) | - | - | - | - | - | - | - | 46.8 | 69.1 | 62.3 | 56.7 | 59.5 |
| Temporal (Fast) | 66.1 | 41.4 | 55.8 | 6.6 | 30.3 | **26.8** | 52.4 | - | - | - | - | - |
| Joint (Slow) | 38.8 | 17.7 | 38.6 | 0.8 | 4.8 | 19.1 | 42.2 | 47.4 | 68.7 | 61.8 | 56.0 | 58.9 |
| Joint (Fast) | 63.3 | 39.0 | 54.3 | 6.5 | 28.6 | 26.2 | 54.8 | 44.6 | 66.2 | 60.0 | 53.6 | 56.8 |
| Joint (Slow/Fast) | 62.2 | 39.0 | 54.0 | 6.0 | 26.4 | 24.2 | 47.1 | 47.6 | 68.9 | 61.6 | 56.0 | 58.8 |
| Joint (SlowFast) | **66.2** | **43.0** | **56.5** | **7.0** | 30.3 | 25.8 | **57.2** | **50.0** | **70.0** | **62.5** | **57.6** | **60.0** |

*Table 6.* Ablation experiments on Training data.

| Dataset | TVG | | VHD | YTVOS | ReVOS | VMME | MME | MMBench | SEED | LoomBench | |
|---|---|---|---|---|---|---|---|---|---|---|---|
| | R1@0.5 | mIoU | mAP | $\mathcal{J}\&\mathcal{F}$ | $\mathcal{J}\&\mathcal{F}$ | Acc | P./R. | Acc | Acc | tIoU | $\mathcal{J}\&\mathcal{F}_{bi\text{-}fore}$ |
| Baseline | 66.2 | 56.5 | 25.8 | 70.0 | 60.0 | 50.7 | 492/115 | 79.0 | 73.9 | 28.1 | 34.6 |
| +VQA | 66.3 | 56.8 | 26.0 | 70.3 | 59.9 | 54.2 | 1684/623 | 80.9 | 74.7 | 29.8 | 36.9 |
| +LoomData | **67.8** | **57.4** | **26.3** | **70.3** | **60.6** | **54.7** | **1699/628** | **81.1** | **75.0** | **34.8** | **41.9** |

*Table 7.* Ablation experiments on Model size and type.

| Backbone | TVG | VHD | ReVOS | LoomBench | |
|---|---|---|---|---|---|
| | mIoU | mAP | $\mathcal{J}\&\mathcal{F}$ | tIoU | $\mathcal{J}\&\mathcal{F}_{bi\text{-}fore}$ |
| InternVL2.5-4B | 57.4 | 26.3 | 60.6 | 34.8 | 41.9 |
| InternVL2.5-8B | 56.4 | 27.1 | 62.0 | 40.2 | 47.2 |
| InternVL3-8B | 59.8 | 27.5 | 63.1 | 41.6 | 49.1 |

that first adopts TimeSuite-7B (Zeng et al., 2025) to localize the relevant clip in the given video, and then applies Sa2VA-8B (Yuan et al., 2025a) to segment the masks based on the user query (denoted as TimeSuite + Sa2VA). We incorporate tIoU and $\mathcal{J}\&\mathcal{F}_{bi\text{-}fore}$ to evaluate *Combined* questions.

As shown in Tab. 4, VideoLoom outperforms the above baseline by a clear margin on *Combined* questions (+16.2 and +15.4 in terms of tIoU and $\mathcal{J}\&\mathcal{F}_{bi\text{-}fore}$). This not only validates the effectiveness of our model on this task, but also underscores the necessity of joint spatial–temporal understanding for comprehensive video comprehension. In addition, we also evaluate VideoLoom on *When* and *Where* questions, demonstrating robust performance in both temporal comprehension and spatial perception.

### 4.3. Ablation Studies

In this section, we conduct extensive ablation experiments using InternVL2.5-4B (Chen et al., 2024), a lightweight MLLM, as our backbone to study the contribution of different components.

**Effects of SlowFast Visual Tokens.** We build different variants to study the effects of SlowFast visual tokens: 1) using only slow tokens to train on spatial tasks, 2) using only fast tokens to train on temporal tasks, 3) using slow or fast tokens and train on both tasks jointly, 4) using fast

tokens for temporal and slow tokens for spatial tasks, and 5) using both slow and fast tokens and train on both tasks. Results of all configurations are compared in Tab. 5.

Using either slow or fast tokens alone leads to substantial performance degradation on spatial or temporal tasks, respectively. The joint (Slow/Fast) setting adopts only fast visual tokens interleaved with timestamps for temporal tasks, and slow visual tokens alone for spatial tasks. It achieves more balanced performance across all datasets, though still with a noticeable drop compared to the specialized single-task models. When SlowFast tokens are employed, the model achieves consistent improvements across nearly all benchmarks, surpassing standalone spatial or temporal models by 4.8 mIoU on QVHighlights and 3.2 $\mathcal{J}\&\mathcal{F}$ on MeVIS. This demonstrates that the proposed SlowFast token design effectively unifies both tasks with mutual benefits and enables coherent spatial–temporal understanding within a single framework.

**Effects of LoomData-8.7K.** Tab. 6 demonstrates the effectiveness of LoomData-8.7K in improving spatial–temporal understanding. We use VideoLoom trained on existing spatial and temporal datasets as the baseline. To eliminate the influence of additional VQA data, we include them only for a fair comparison. The results show that with LoomData-8.7K, our model achieves an improvement of +5.0 $\mathcal{J}\&\mathcal{F}_{bi\text{-}fore}$ in joint spatial–temporal understanding, along with consistent gains across all benchmarks, including spatial, temporal, and general visual comprehension (VideoMME (Fu et al., 2025b), MME (Fu et al., 2025a), MMBench (Liu et al., 2024c), and SEED-Bench (Li et al., 2024a)). These results demonstrate that LoomData-8.7K could provide high-quality supervision for joint spatial–temporal understanding, with consistent spatial trajectories and temporal annotations.

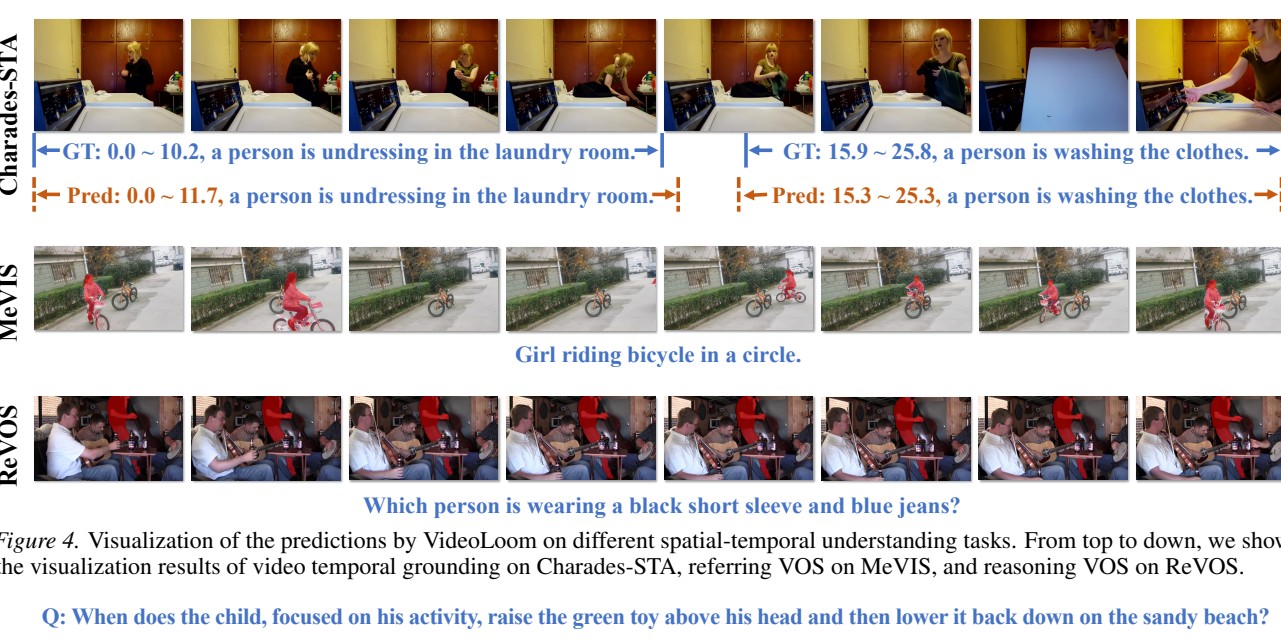

*Figure 4.* Visualization of the predictions by VideoLoom on different spatial-temporal understanding tasks. From top to down, we show the visualization results of video temporal grounding on Charades-STA, referring VOS on MeVIS, and reasoning VOS on ReVOS.

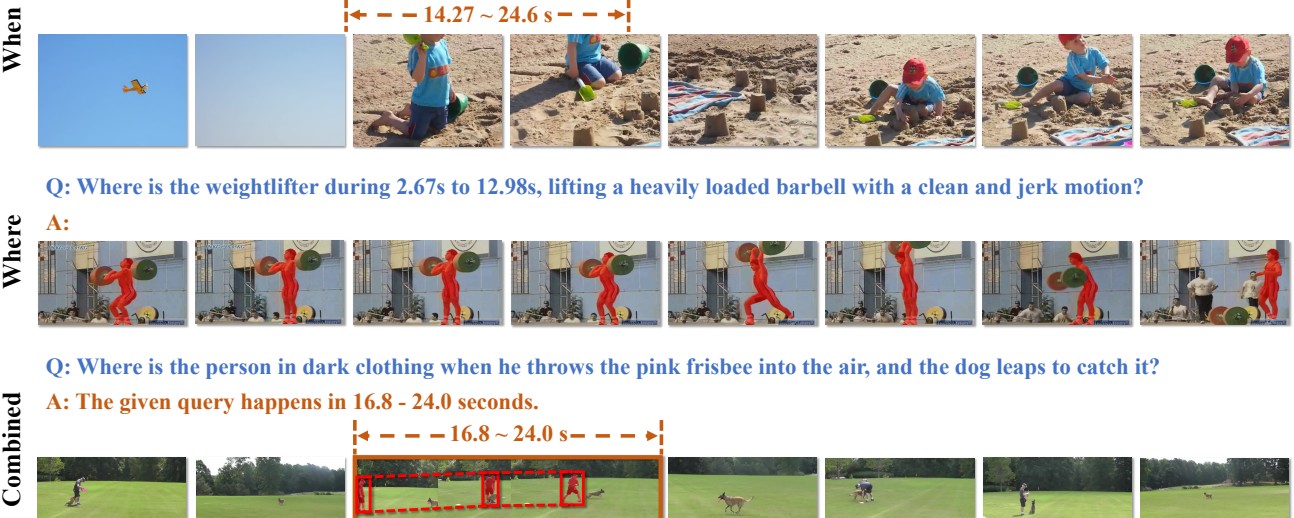

*Figure 5.* Visualization of VideoLoom on LoomBench for *When*, *Where*, and *Combined* questions.

**Effects of Base Models.** To evaluate the impact of different base models on spatial–temporal understanding, we conduct experiments using various MLLMs as backbones. As shown in Tab. 7, VideoLoom achieves higher performance with InternVL2.5-8B (Chen et al., 2024) compared to its smaller InternVL2.5-4B counterpart, indicating that larger MLLMs provide stronger multimodal representations for spatial–temporal reasoning. When equipped with the more advanced InternVL3-8B (Zhu et al., 2025), further improvements are observed under comparable model capacities. These results demonstrate that VideoLoom continually benefits from advancements in underlying MLLMs, showing strong scalability and the potential for enhanced spatial–temporal understanding as foundation models evolve.

### 4.4. Visualizations

**Qualitative Cases across Multiple Benchmarks.** We present qualitative visualizations of VideoLoom across multiple spatial–temporal understanding datasets in Fig. 4. The first row illustrates that our model accurately localizes events along the temporal dimension. The following two rows show its capability to perform object segmentation conditioned on diverse types of textual references.

Additionally, Fig. 5 provides qualitative examples across the three question types from LoomBench, further illustrating the strong joint spatial–temporal understanding capability of VideoLoom. For instance, in the query "Where is the person in dark clothing when he throws the pink frisbee

**Where is the man in the white graphic t-shirt and blue jeans when he jumps on the mini-trampoline, raising his arms rhythmically as he exercises?**
TimeSuite + Sa2VA pipeline: 36.0 - 111.5s

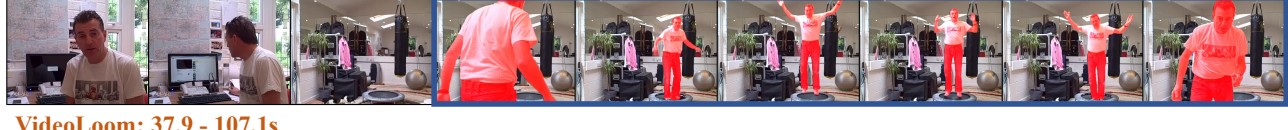

VideoLoom: 37.9 - 107.1s

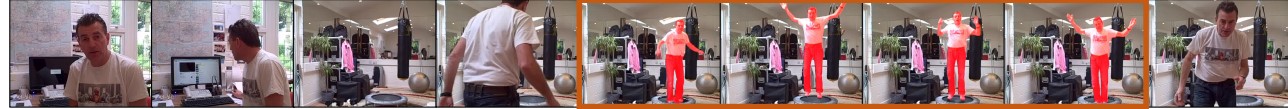

**Where is the cowboy when they ride a horse and chase a white calf, holding onto a rope?**
TimeSuite + Sa2VA pipeline: 0.0 - 10.0s

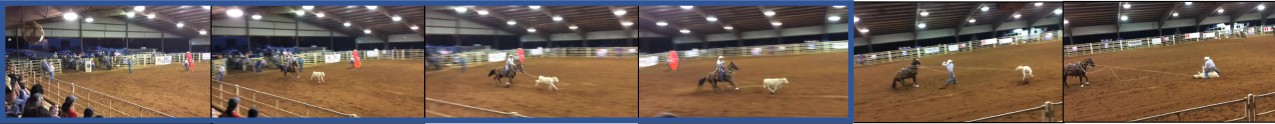

VideoLoom: 3.0 - 9.5s

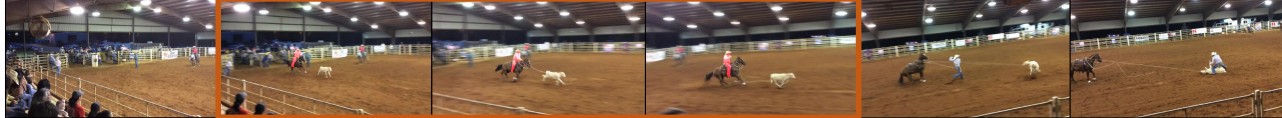

*Figure 6.* Visual comparison between VideoLoom and the TimeSuite+Sa2VA pipeline, illustrating bidirectional information exchange in joint spatial–temporal modeling.

into the air, and the dog leaps to catch it", VideoLoom first localizes the relevant temporal segment corresponding to the throwing action and then accurately identifies the spatial region of the person within that interval. This example demonstrates its ability to reason across both time and space, linking dynamic actions to precise spatial localization within a unified framework.

**Visual Analysis of Joint Spatial–Temporal Modeling.** To further elaborate on the necessity and merits of joint spatial–temporal modeling, we present qualitative cases in Fig. 6 to explore the complementary information exchanged between spatial and temporal reasoning.

Spatial and temporal understanding mutually reinforce each other in a bidirectional manner. Spatial perception aggregates high-level semantics into the `[SEG]` token, enhancing target awareness. In the first row, VideoLoom leverages spatial cues (the spatial relationship between "the man" and "the mini-trampoline") to refine temporal localization, identifying more precise action boundaries than the disjoint pipeline. Conversely, temporal grounding aligns queries with relevant frames, providing coarse-to-fine guidance for region-level identification, facilitating precise segmentation. In the second row, while both models locate similar intervals, only VideoLoom accurately identifies the correct "cowboy" in high-motion scenes, demonstrating superior semantic alignment. These results demonstrate that joint modeling enables coherent spatial-temporal understanding

by facilitating bidirectional information exchange. By fully exploiting the complementarity between space and time, VideoLoom unlocks spatial-temporal capabilities that fundamentally surpass the limitations of compositional pipelines.

## 5. Conclusion

This work presents the VideoLoom suite to advance joint spatial-temporal understanding for the first time. It comprises three key components: 1) LoomData-8.7k, a character-centric dataset that provides both timestamp-aligned action descriptions and fine-grained spatial masks. 2) VideoLoom, a unified Video LLM equipped with MLLM-SAM2 architecture and SlowFast visual tokens to generate both temporal locations and spatial masks. and 3) LoomBench, a novel benchmark designed to evaluate Video LLMs across diverse question types, *When*, *Where*, and *Combined*, for a comprehensive assessment of spatial-temporal understanding. Extensive experiments on a range of spatial and temporal benchmarks demonstrate that VideoLoom achieves strong performance and establishes new state-of-the-art results in time and space. We hope our VideoLoom could shed new light on unified Video LLMs.

## Acknowledgments

This project was supported by NSFC under Grant No. 62472098 and No. 624B2043.

## Impact Statement

This paper presents work whose goal is to advance the field of Machine Learning. There are many potential societal consequences of our work, none which we feel must be specifically highlighted here.

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

# Appendix

## A. Overview

Our supplementary includes the following sections:

- **Sec. B: Model details.** Details for VideoLoom design, implementation and training data.

- **Sec. C: LoomData details.** Details for manual verification, and statistics for LoomData-8.7k.

- **Sec. D: More experiment results.** Analysis on Bidirectional Foreground $\mathcal{J}\&\mathcal{F}$, and additional performance evaluation.

- **Sec. E: More visualization.** More visualization of our dataset and results.

- **Sec. F: Prompt design.** Prompt for temporal action annotation and LoomBench construction.

- **Sec. G: Limitations.** Shortcomings and future directions of VideoLoom suite.

## B. Model Details

### B.1. More Details about VideoLoom

**Interleaved Input.** For temporal modeling, we interleave temporal information, i.e., unique frame IDs, with fast visual tokens. Specifically, we insert frame IDs, e.g., "This sampled frame id is 26", after the fast tokens of the corresponding frames, leading to an interleaved sequence. We then concatenate this token sequence with the slow tokens as input $I$ to the LLM:

$$I = [\mathrm{F}_1; \mathrm{ID}_1; ...; \mathrm{F}_{N_f}; \mathrm{ID}_{N_f}; \mathrm{S}_1; ...; \mathrm{S}_{N_s}]. \tag{4}$$

where $\mathrm{ID}_j$, $\mathrm{F}_j$, $\mathrm{S}_k$ denote the ID text tokens, fast tokens, and slow tokens, while $N_f$ and $N_s$ for the count of frames with fast and slow tokens.

By directly using numerical text of frame IDs to represent temporal positions, temporal understanding is transformed into language instruction QA, aligning with the general capabilities of MLLMs.

**[SEG] token.** To generate masks for keyframes, SAM2 (Ravi et al., 2025) only needs to activate a visual encoder and a mask decoder. Given the keyframes sampled for slow tokens, we extract visual features $f_v$ using the visual encoder, which provides pixel-level details for trajectory prediction. The SAM2 mask decoder is connected to MLLM via a [SEG] token contained in the text output. Since MLLM performs fine-grained spatial-temporal modeling with SlowFast tokens, the [SEG] token captures rich target information under segmentation queries. The hidden states of the [SEG] token, denoted as $h_{seg}$, pass through an MLP projection layer to form a target embedding. This embedding serves as a novel visual prompt for SAM2, fed into the mask decoder with the visual features $f_v$ to generate masks $M_v$ for the keyframes:

$$M_v = \mathrm{SAM2}(f_v, \mathrm{MLP}(h_{\mathrm{seg}})). \tag{5}$$

### B.2. Additional Implemental Details

Tab. 8 lists hyperparameters for one-stage tuning. Specifically, for the number of frames for fast tokens, we adopt different settings across datasets based on video duration, with a maximum of 128 frames. For Charades-STA (Gao et al., 2017), where videos typically last around 30 seconds, we sample 64 frames for fast tokens. For YouCook2 (Zhou et al., 2018), where videos often exceed 2 minutes in length, we uniformly sample 128 frames. For QVHighlights (Lei et al., 2021), annotated in 2-second intervals, we sample frames at 2 FPS, typically yielding around 75 frames. For spatial datasets (Seo et al., 2020; Ding et al., 2023; Yan et al., 2024), which provide annotated frame sequences, we uniformly sample up to 64 frames.

### B.3. Training Data

We present all the datasets for training and report their item counts and repeat times in Tab. 9. Finally, VideoLoom is jointly trained for 1,315K iterations and achieves advanced performance on all these tasks.

*Table 8.* Hyperparameters for one-stage tuning.

| Hyperparameter | Value |
|---|---|
| Epochs | 1 |
| Batch size | 64 |
| Learning rate | 4e-5 |
| Weight decay | 0.05 |
| AdamW $\beta$ | (0.9, 0.999) |
| Max sequence length for MLLM | 8192 |
| Number of fast tokens per frame | 16 |
| Number of slow tokens per frame | 256 |
| Frame resolution for MLLM | $448 \times 448$ |
| Frame resolution for SAM2 | $1024 \times 1024$ |
| Number of frames for fast tokens | $\leq 128$ |
| Number of frames for slow tokens | 5 |

*Table 9.* Training datasets, item counts, and repeat times.

| Dataset | Item count | Repeats |
|---|---|---|
| LLaVA (Liu et al., 2024a) | 665K | 1 |
| RefCOCO (Kazemzadeh et al., 2014) | 17K | 4 |
| RefCOCO+ (Kazemzadeh et al., 2014) | 17K | 4 |
| RefCOCOg (Yu et al., 2016) | 17K | 4 |
| Grand-f (Rasheed et al., 2024) (Auto) | 196K | 1 |
| Grand-f (Rasheed et al., 2024) (Human) | 1K | 10 |
| Charades-STA (Gao et al., 2017) | 12.4K | 4 |
| YouCook2 (Zhou et al., 2018) | 1.2K | 10 |
| QVHighlights (Lei et al., 2021) | 6.9K | 4 |
| **LoomData for VTG** | 8.7K | 4 |
| Ref-YTVOS (Seo et al., 2020) | 3.5K | 12 |
| MeVIS (Ding et al., 2023) | 1.6K | 12 |
| ReVOS (Yan et al., 2024) | 1.7K | 12 |
| **LoomData for refVOS** | 8.7K | 4 |

## C. LoomData Details

### C.1. Details about Manual Verification

Here we introduce the simple manual verification process in the pipeline, explaining how to implement filtering and correction of complete tracklets after spatial mask annotation. This approach involves two rounds of simple judgments:

In the first round, we primarily focus on filtering out videos with missing annotations, as completing the missing tracklets requires extensive manual annotation, which is not scalable. Specifically, we display the annotation on the middle frame of the longest shot (i.e., the key frame where we initially identify the main character) as a reference, and then display the middle frames of the shots without tracklets in turn. We then manually determine whether there is an unlabeled main character in these frames, discarding the video if one is found, as shown in Fig. 7. (i).

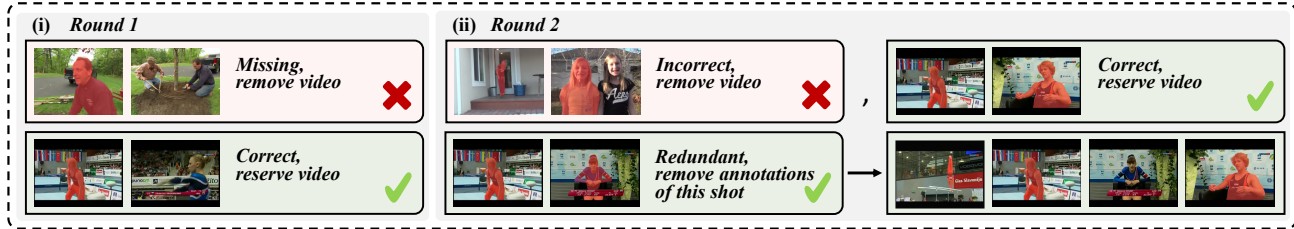

*Figure 7.* Examples of manual verification. (i) In the first round, we filter out videos with missing annotations. (ii) In the second round, we filter out videos with incorrect annotations and remove redundant annotations from the shots of the retained videos.

The second round of verification focuses on the shots with tracklets, where we filter out videos with incorrect annotations and remove redundant annotations from the retained shots. For a shot with tracklets, we define incorrect annotations as the presence of the main character but the mask labeled to other objects, and redundant annotations as the absence of the main character but the mask labeled to other objects. We discard entire videos containing incorrectly labeled shots and remove annotations from redundantly labeled shots to make a simple revision of the video, as shown in Fig. 7. (ii). Specifically, we continue to display the annotation of the middle frame of the longest shot for reference purposes and display the middle frames of the shots with tracklets in turn. We then manually determine whether the annotations on these frames are incorrect, redundant, or correct to carry out the corresponding operations.

To construct LoomData, we selected 2,735 videos from ActivityNet (Caba Heilbron et al., 2015) for spatial mask annotation. After the first round, 743 missing videos were filtered out. 380 incorrect videos were discarded, and 461 redundant videos were rectified in the second round, resulting in a total of 1,612 videos for subsequent procedures. Consequently, this automatic annotation pipeline achieves a first-pass accuracy of 42.1%, with 58.9% of videos accepted after verification. And the manual verification process requires minimal human input, with a single annotator completing it in approximately 25 hours. This demonstrates the quality control and scalability of the spatial-temporal annotation pipeline.

## C.2. Automatic Filtering Rules

During Shot Partition, we refine the timestamp boundaries by merging clips shorter than 1 second, which filters out fragmented segments and camera instability. Furthermore, to maintain annotation efficiency and minimize complexity, we focus on videos with a moderate number of transitions, excluding those containing more than 10 shots.

After Spatial Mask Annotation, manually verified videos accepted by manual verification undergo a further filtering process. We keep videos only if: (1) the proportion of labeled frames exceeds a specific threshold $\tau_{prop}$, ensuring temporal continuity, and (2) the average mask area per frame remains below another threshold $\tau_{area}$, preventing trivial cases where the target occludes the majority of the background. $\tau_{prop}$ and $\tau_{area}$ are set to 0.4 and 0.5, respectively.

To ensure the quality of Temporal Action Annotation, we employ BLIP (Li et al., 2022) to evaluate the semantic similarity, following HowToCaption (Shvetsova et al., 2024). Specifically, we calculate the similarity between each description and its corresponding video shot to ensure cross-modal alignment. Simultaneously, we measure the similarity among all descriptions to ensure that different action labels are sufficiently discriminative. Only videos satisfying both high alignment and high distinctiveness are preserved.

## C.3. Statistics for LoomData-8.7k

Tab. 10 compares our constructed dataset, LoomData-8.7k, with existing spatial-temporal datasets. For the first time, LoomData achieves joint annotation of temporal timestamps and spatial masks on nearly 2-minute videos. LoomData enables fine-grained temporal partition, with each video containing an average of 6.0 segments with tracklets, comparable to current spatial-temporal datasets. Compared to temporal datasets, which only roughly label overlapping temporal locations, LoomData achieves a complete temporal partition of the videos while providing a more detailed description. Compared to spatial datasets, LoomData achieves mask-level annotation while ensuring instance consistency across the entire video. Fig. 8 shows the distribution of shot lengths and normalized shot center timestamps (by video duration). LoomData contains shots of widely varying lengths. Over 50% of shots are concentrated in the range from 5 to 15 seconds, while only a few exceed 30 seconds. These shots are almost evenly distributed across the videos, suggesting that LoomData suffers less from temporal bias.

*Table 10.* Comparison with existing spatial-temporal datasets.

| Dataset | #Videos | Avg #Segments | Avg #Tracklets | Avg Len (sec) Segment/Video | Temporal Ann. | Box Ann. | Mask Ann. |
|---|---|---|---|---|---|---|---|
| Charades-STA (Gao et al., 2017) | 5,338 | 6.8 | - | 8.1/30.6 | ✓ | | |
| ANet Captions (Krishna et al., 2017) | 10,024 | 3.7 | - | 36.2/117.6 | ✓ | | |
| RefYTVOS (Seo et al., 2020) | 3,471 | - | 1.9 | - | | | ✓ |
| MeVIS (Ding et al., 2023) | 1,662 | - | 4.3 | - | | | ✓ |
| VidSTG (Zhang et al., 2020) | 5,563 | 6.5 | 5.0 | 9.7/28.0 | ✓ | ✓ | |
| TVQA+ (Lei et al., 2020) | 3,364 | 7.0 | 19.4 | 7.2/61.5 | ✓ | ✓ | |
| LoomData | 1,456 | 6.0 | 6.0 | 15.0/102.2 | ✓ | | ✓ |

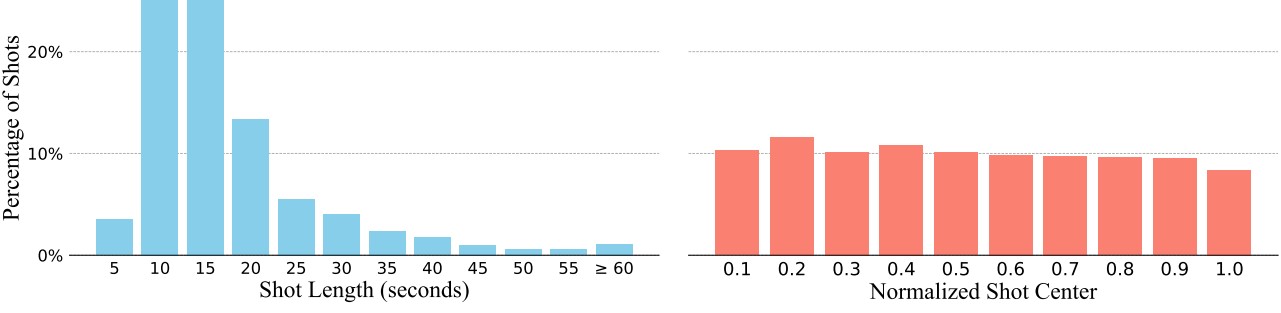

*Figure 8.* Distribution of shot lengths (left) and normalized (by video duration) center timestamps (right). The shots vary widely in length, and they distribute almost evenly along the videos.

## C.4. Quality Control for LoomBench

After manual verification and rule-based filtering, we obtain the joint annotations for LoomBench construction. To mitigate error accumulation, we manually refine preliminary annotations before question generation and correct any errors generated by LLM. This process ensures consistent annotated descriptions and accurate temporal boundaries and spatial masks, while effectively eliminating hallucinations in questions. The entire curation effort for LoomBench exceeded 50 person-hours, guaranteeing the high quality of fine-grained spatial-temporal QA pairs.

# D. More Experiment Results

## D.1. Analysis on Bidirectional Foreground J&F

We propose a new evaluation metric, Bidirectional Foreground $\mathcal{J}\&\mathcal{F}$, for assessing the joint spatial-temporal understanding of *Combined* questions on LoomBench. In this section, we first demonstrate the necessity with experimental results under varying queried segment lengths. We then present the specific values of each component of this metric to provide an in-depth assessment.

**The Necessity of $\mathcal{J}\&\mathcal{F}_{bi\text{-}fore}$.** The *Combined* questions are divided by the percentage of length of the queried segment over the entire video, into three categories: 0-20%, 20-60%, and 60-100%. We then provide a comparative analysis of the standard $\mathcal{J}\&\mathcal{F}$ and our proposed $\mathcal{J}\&\mathcal{F}_{bi\text{-}fore}$ in Tab. 11.

*Table 11*. Comparison between standard $\mathcal{J}\&\mathcal{F}$ and $\mathcal{J}\&\mathcal{F}_{bi\text{-}fore}$ on *Combined* questions of LoomBench, under varying queried segment lengths.

| Metric | 0-20% | 20-60% | 60-100% | All |
|---|---|---|---|---|
| Standard $\mathcal{J}\&\mathcal{F}$ | 88.9 | 77.7 | 41.0 | 83.3 |
| $\mathcal{J}\&\mathcal{F}_{bi\text{-}fore}$ | 47.6 | 50.8 | 37.1 | 49.1 |

We can see that $\mathcal{J}\&\mathcal{F}_{bi\text{-}fore}$ performs stably under variable-length queried segments, while the standard $\mathcal{J}\&\mathcal{F}$ increases significantly with shorter lengths, resulting in a substantial gap between 0-20% and 60-100%. This discrepancy stems from the calculation $\mathcal{J}\&\mathcal{F} = \frac{\bar{\mathcal{J}}+\bar{\mathcal{F}}}{2}$, where $\bar{\mathcal{J}}$ and $\bar{\mathcal{F}}$ denote the average Jaccard index and F-score across all video frames, respectively. For frames without groundtruth masks, i.e., background frames, when the predicted mask is None, the value of $\mathcal{J}$ and $\mathcal{F}$ both reach 1 (100%). When averaged over the entire video, the standard metric is significantly influenced by the easily predicted background frames, leading to inflated values and excessive sensitivity to the proportion of foreground queries, which prevents a correct assessment of spatial-temporal capabilities.

Our proposed Bidirectional Foreground $\mathcal{J}\&\mathcal{F}$ metric decouples the assessment into two critical dimensions, i.e., $\mathcal{J}\&\mathcal{F}_g$ computed over the temporal span of groundtruth masklet and $\mathcal{J}\&\mathcal{F}_p$ computed over the temporal span of predicted masklet. The former assesses spatial recall during character presence, and the latter evaluates temporal precision and reliability. By employing a harmonic mean, $\mathcal{J}\&\mathcal{F}_{bi\text{-}fore}$ balances these two dimensions and effectively filters out the noise of trivial 100% scores from background frames, providing a robust and high-fidelity evaluation. The formal derivation of Eq. 3 is presented below:

$$
\begin{aligned}
\mathcal{J}\&\mathcal{F}_{bi\text{-}fore} &= \frac{2 \cdot \mathcal{J}\&\mathcal{F}_p \cdot \mathcal{J}\&\mathcal{F}_g}{\mathcal{J}\&\mathcal{F}_p + \mathcal{J}\&\mathcal{F}_g} \\
&= \frac{2 \cdot \left(\frac{\mathcal{J}_p+\mathcal{F}_p}{2}\right) \cdot \left(\frac{\mathcal{J}_g+\mathcal{F}_g}{2}\right)}{\frac{\mathcal{J}_p+\mathcal{F}_p}{2} + \frac{\mathcal{J}_g+\mathcal{F}_g}{2}} \\
&= \frac{(\mathcal{J}_p + \mathcal{F}_p) \cdot (\mathcal{J}_g + \mathcal{F}_g)}{(\mathcal{J}_p + \mathcal{F}_p) + (\mathcal{J}_g + \mathcal{F}_g)}
\end{aligned}
\tag{6}
$$

where $\mathcal{J}_p$ and $\mathcal{F}_p$ denote the average Jaccard index and F-score computed over the predicted temporal span, while $\mathcal{J}_g$ and $\mathcal{F}_g$ are the corresponding metrics averaged over the groundtruth span. Specifically, the component metrics are formally defined as follows:

$$
\begin{aligned}
\mathcal{J}_p &= \mathcal{J}_{\text{Loc}(P)}(P, G), & \mathcal{F}_p &= \mathcal{F}_{\text{Loc}(P)}(P, G) \\
\mathcal{J}_g &= \mathcal{J}_{\text{Loc}(G)}(P, G), & \mathcal{F}_g &= \mathcal{F}_{\text{Loc}(G)}(P, G)
\end{aligned}
\tag{7}
$$

where $P$ and $G$ denote the sequences of predicted and groundtruth masks, respectively. The function $\text{Loc}(\cdot)$ extracts the continuous temporal range of a masklet, spanning from the first to the last frame where a valid mask is present, regardless of intermediate occlusions:

$$
\text{Loc}(\mathbf{m}) = \{t \mid \min(T_\mathbf{m}) \leq t \leq \max(T_\mathbf{m})\}
\tag{8}
$$

where $T_{\mathbf{m}} = \{t \mid \mathbf{m}_t \neq \emptyset\}$ is the set of indices for frames containing valid masks.

Referring VOS (Seo et al., 2020; Ding et al., 2023; Yan et al., 2024) adopts standard $\mathcal{J}\&\mathcal{F}$ as evaluation metrics because videos in existing datasets are often foreground throughout (up to 60% or more, as indicated by Tab. 11 showing close values for the two metrics on segments with length of 60-100%), which is notably different from LoomBench. To effectively evaluate performance on LoomBench, we utilize $\mathcal{J}\&\mathcal{F}_{bi\text{-}fore}$, thereby avoiding extensive computation on background frames and ensuring accurate assessment for spatial-temporal comprehension.

**In-depth Comparison of Components.** We present the specific values of each component of $\mathcal{J}\&\mathcal{F}_{bi\text{-}fore}$ in Tab. 12, including $\mathcal{J}_p$, $\mathcal{F}_p$, $\mathcal{J}\&\mathcal{F}_p$ computed over the predicted masklet, and $\mathcal{J}_g$, $\mathcal{F}_g$, $\mathcal{J}\&\mathcal{F}_g$ computed over the groundtruth. The experimental results demonstrate that VideoLoom outperforms the baseline, which consists of TimeSuite (Zeng et al., 2025) and Sa2VA (Yuan et al., 2025a), across all metrics. Additionally, it is evident that the metric scores computed over the predicted masklet are higher than those computed over the groundtruth, highlighting the superior precision of the model predictions, though a notable gap remains in recall.

*Table 12.* Detailed results of VideoLoom on *Combined* questions of LoomBench.

| Method | $\mathcal{J}_p$ | $\mathcal{F}_p$ | $\mathcal{J}\&\mathcal{F}_p$ | $\mathcal{J}_g$ | $\mathcal{F}_g$ | $\mathcal{J}\&\mathcal{F}_g$ | $\mathcal{J}\&\mathcal{F}_{bi\text{-}fore}$ |
|---|---|---|---|---|---|---|---|
| TimeSuite+Sa2VA | 47.0 | 48.9 | 48.0 | 25.4 | 26.6 | 26.0 | 33.7 |
| VideoLoom-8B | 58.1 | 60.5 | 59.3 | 41.1 | 42.8 | 41.9 | 49.1 |

### D.2. Ablation on Non-Human Categories

To demonstrate the generalizability of VideoLoom on human and non-human categories, we conduct ablation experiments on RefDavis17 (Khoreva et al., 2019), a benchmark for referring VOS, in a zero-shot setting. We divide the classes of objects and report the results separately in Tab. 13.

*Table 13.* Ablation experiments on Human and Non-human categories of RefDavis17.

| Method | Human | | | Non-Human | | | All | | |
|---|---|---|---|---|---|---|---|---|---|
| | $\mathcal{J}$ | $\mathcal{F}$ | $\mathcal{J}\&\mathcal{F}$ | $\mathcal{J}$ | $\mathcal{F}$ | $\mathcal{J}\&\mathcal{F}$ | $\mathcal{J}$ | $\mathcal{F}$ | $\mathcal{J}\&\mathcal{F}$ |
| w/o LoomData | 73.0 | 82.0 | 77.5 | 63.1 | 72.3 | 67.7 | 67.5 | 76.6 | 72.1 |
| Ours | 75.4 | 84.3 | 79.8 | 65.7 | 74.3 | 70.0 | 70.0 | 78.7 | 74.3 |

With or without LoomData, the performance of segmentation on human class surpasses that of the non-human classes. Moreover, incorporating our constructed LoomData leads to a notable enhancement in the segmentation of the human class (+2.3 $\mathcal{J}\&\mathcal{F}$), while also benefiting the segmentation of non-human classes (+2.3 $\mathcal{J}\&\mathcal{F}$). This suggests that, although our data primarily targets the human class, detailed textual descriptions contribute to the comprehension of semantics across various categories. Consequently, VideoLoom demonstrates the ability to generalize to any category and greatly benefits from the human class annotations provided by LoomData.

### D.3. Ablation on Multi-Entity Interactions

Although LoomData focuses on a single main character, its annotations are not limited to single-entity descriptions. Many samples explicitly involve interactions between multiple entities, e.g., "The man lunges forward to engage with a woman, grappling with her. He quickly spins her around before bending at the waist." This means LoomData still provides supervision for relational understanding in multi-entity scenarios.

To quantitatively assess this capability, we further select 275 interaction-centric queries from MeVIS (Ding et al., 2023), such as "rabbit leaping over another rabbit", and conduct an ablation study in Tab. 14.

The results show that incorporating LoomData brings a notable improvement of +1.6 $\mathcal{J}\&\mathcal{F}$ on these interaction queries. It demonstrates that LoomData provides useful supervision not only for single-entity localization but also for relational reasoning in multi-entity interaction scenarios.

*Table 14.* Ablation experiments on Multi-entity Interactions of MeVIS.

| Method | $\mathcal{J}$ | $\mathcal{F}$ | $\mathcal{J}\&\mathcal{F}$ |
|---|---|---|---|
| w/o LoomData | 59.1 | 68.2 | 63.6 |
| Ours | 60.9 | 69.5 | 65.2 |

### D.4. Detailed Comparison with Sa2VA

We conduct a fair comparison with Sa2VA (Yuan et al., 2025a), the model most closely aligned with our approach. Following

Sa2VA, we employ InternVL2.5-4B (Chen et al., 2024) as the MLLM backbone to report our results in Tab. 15. We can see that VideoLoom significantly surpasses Sa2VA on MeVIS (Ding et al., 2023) and also achieves competitive performance on RefYTVOS (Seo et al., 2020), highlighting its superior motion capture and reasoning capabilities.

*Table 15.* Comparison with Sa2VA using the same backbone.

| Method | Backbone | MeVIS_u | MeVIS | YTVOS |
|---|---|---|---|---|
| | | $\mathcal{J}\&\mathcal{F}$ | $\mathcal{J}\&\mathcal{F}$ | $\mathcal{J}\&\mathcal{F}$ |
| Sa2VA (Yuan et al., 2025a) | InternVL2.5-4B | 55.9 | 46.4 | 71.3 |
| VideoLoom | InternVL2.5-4B | 60.9 | 50.6 | 70.3 |

### D.5. Computational Efficiency

To evaluate the efficiency of VideoLoom, we report the computational metrics for our 4B model in Tab. 16, using a 64-frame input for fast tokens. For temporal tasks, the model only involves the MLLM architecture, achieving an ultra-fast average

response of 0.25s. For spatial tasks, although the additional activation of the SAM2 branch introduces mask decoding overhead, the model maintains a high inference speed of 0.62s, making it suitable for real-time applications. Notably, the peak GPU memory remains below 14 GB across both tasks, facilitating deployment on a single 16 GB GPU. These results demonstrate that VideoLoom effectively integrates MLLM and SAM2, providing high responsiveness with modest computational costs.

*Table 16.* Comparison of the Computational Efficiency.

| Task | FLOPS | Peak GPU Memory | Inference Time on single H20 GPU |
|---|---|---|---|
| Temporal | 22.4 T | 13.4 GB | 0.25 s |
| Spatial | 31.7 T | 13.8 GB | 0.62 s |

## E. More Visualization

### E.1. Visualization of Full Annotation

To visualize the annotation results of our pipeline, we present an example of the complete spatial-temporal annotation for a randomly selected video in Fig. 9. This annotation fully captures the timestamp-aligned actions and mask-level locations of the main characters.

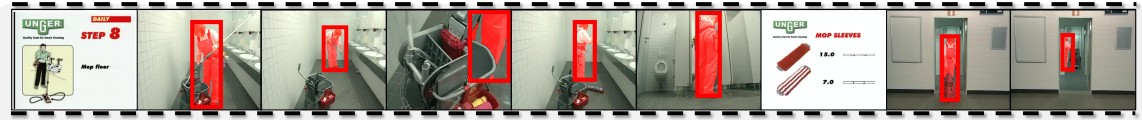

**6.0s - 19.0s:** A person wearing grey overalls, a white shirt, and gloves stands by a cleaning cart. They take a long-handled mop from the cart. They then attach a red mop head to the bottom of the pole, preparing it for use.

**19.0s - 48.5s:** The person begins mopping the tiled floor in a back-and-forth motion. They start near the row of sinks and work their way backwards, away from the cleaning cart. They maintain a slightly bent posture while mopping.

**48.5s - 66.44s:** The person returns to the cleaning cart and dunks the mop head into the bucket. They place the wet mop into the attached press wringer. They then operate the wringer with the mop handle to squeeze out the water.

**66.44s - 90.64s:** The person lifts the mop, and after the head briefly detaches and is reattached, they resume mopping. They continue cleaning the area near the sinks before moving over to mop the floor around the base of the toilet stalls.

**120.64s - 128.64s:** A person wearing grey overalls over a white shirt walks out from a hallway. They bend down to pick up a red safety cone from the floor. Holding the cone, the person turns around and begins walking back down the hallway.

**128.64s - 136.0s:** The person walks a few more steps and stops beside a cleaning cart. They turn towards the cart and remain stationary, appearing to arrange items on it.

*Figure 9.* An example of the complete spatial-temporal annotation of a video.

### E.2. Qualitative Results and Failure Cases

We present additional qualitative results of VideoLoom across multiple spatial-temporal tasks. As illustrated in Fig. 10, VideoLoom can follow diverse spatial-temporal instructions and establish a solid baseline across different tasks. However, in complex joint understanding scenarios (e.g., when querying sub-actions or the *n*-th occurrence), it occasionally generates

inaccurate spatial-temporal locations, as shown in Fig. 11. This issue likely arises from limitations in temporal action grounding. When confronted with lengthy queries that require complex reasoning across both spatial and temporal dimensions, the model struggles to identify complete temporal intervals spanning the entire motion sequence, which may lead to misaligned spatial-temporal localization.

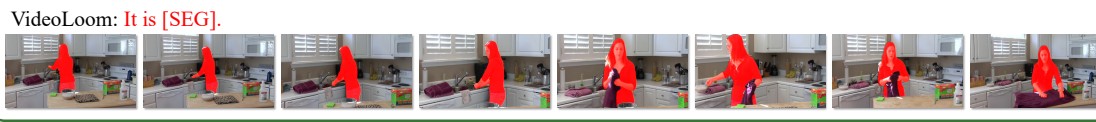

*Figure 10.* Additional qualitative results of VideoLoom on diverse spatial-temporal tasks.

Where is the woman cleaning the red boot when she wipes the surface, straps, and heel with a damp cloth?

**GT**: 49.84 - 75.38s

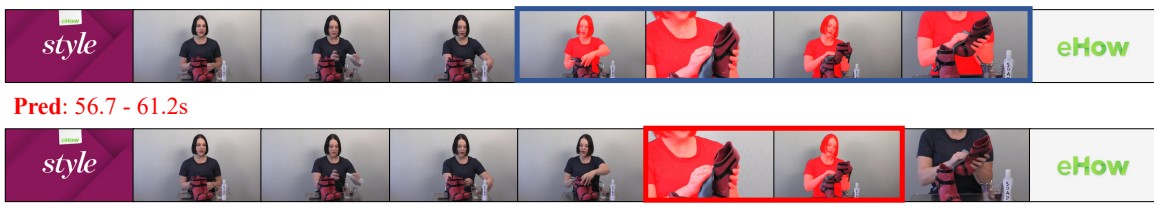

**Pred**: 56.7 - 61.2s

- - - - - - - - - - - - - - - - - - - - - - - - - - - - - - - - - - - - - - - - - - - - - - - - - - - -

Where is the woman in a purple and white leotard when she performs the second roll on her back, twirling the baton between her feet and then rises to a standing position?

**GT**: 105.5 - 113.0s

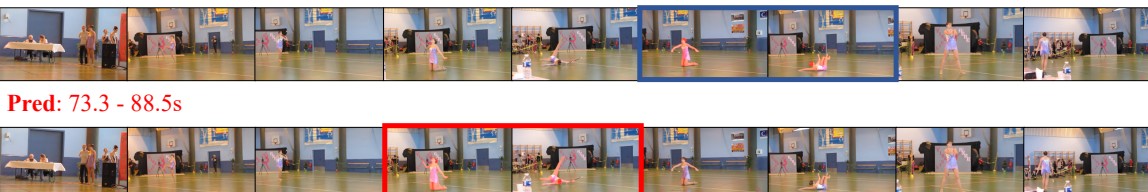

**Pred**: 73.3 - 88.5s

*Figure 11.* Failure cases of VideoLoom on LoomBench, e.g., when querying sub-actions or the *n*-th occurrence.

We intend to mitigate this limitation in future work by enhancing the reasoning process. Specifically, we plan to introduce Chain-of-Thought (CoT) reasoning for temporal grounding, following Time-R1 (Wang et al., 2026b), and further boost reasoning performance via reinforcement learning.

## F. Prompt Design

### F.1. Prompt for Temporal Action Annotation

Prompt engineering plays a vital role in guiding Gemini2.5pro (Comanici et al., 2025) to generate detailed and specific action descriptions aligned with frame IDs for video shots. The prompt utilized is illustrated in Fig. 12. To ensure clarity and precision, we first outline the task of generating instance-level descriptions of actions and appearances using visual prompts from SoM (Yang et al., 2023) and NumPro (Wu et al., 2025). The former annotates instance-level IDs on the main character, while the latter sequentially labels unique frame IDs on each frame. Next, we provide a series of instructions, including Frame Range Division, Description Content, Writing Style, and Output Format. These guidelines ensure concise, distinct, and formatted output with complete temporal coverage, avoiding irrelevant descriptions. Finally, we provide an example output and specify the number of sampled frames for the shot, ensuring alignment between the descriptions and the frame IDs. As a result, Gemini2.5pro generates clear and accurate instance-level descriptions of the main character based on the visual content of the current shot, under the guidance of our carefully designed prompt.

### F.2. Prompt for LoomBench Construction

We prompt LLaMA3.1 (Grattafiori et al., 2024) to generate *When*, *Where*, and *Combined* questions based on annotations produced by our pipeline, and we show the prompt in Fig. 13. We first define the task to create detailed and context-aware questions from video shot descriptions explicitly. Next, we specify the requirements for each of the three question types, emphasizing that timestamps should not appear in *Combined* or *When* questions. Finally, we present a concrete example to clarify the form of the questions further. Based on each shot description, LLaMA3.1 subsequently generates three categories of questions, both detailed and context-aware, to construct the LoomBench.

## G. Limitations

While significantly reducing manual effort and enabling scalable annotation, our proposed annotation pipeline still involves multiple stages with interdependent components. Moreover, the current framework lacks an explicit feedback loop for the MLLM when SAM2 fails to generate valid segmentation masks. In the future, we plan to further automate this pipeline by integrating stronger multimodal foundation models and agents for both annotation generation and verification, aiming

to improve efficiency and expand the annotation scale and categories. Beyond that, we intend to introduce a reflection mechanism after mask generation. The model first evaluates the validity of predicted masks via rule-based criteria, external instance-understanding models, or a rejection token, and then recursively refines temporal localization based on such feedback. If the mask is only partially valid, the interval can be refined; if it is invalid throughout, the segment can be excluded, and grounding can be re-run.

**Task Overview:**
You are given a video segment consisting of a sequence of frames, where each frame is marked with a red numeric frame ID in the lower left corner indicating its sequential order. In each frame, the main person is labeled with a bright numeric ID "1" at their center and boundary.
Your task is to generate detailed, instance-level descriptions of actions and appearance of the main person for the entire video segment, dividing the video frames into contiguous, non-overlapping frame ranges based on significant changes in the actions or movements. Follow these instructions carefully:

**Instructions for writing the detailed description:**
Frame Range Division:
1. Divide the video frames into contiguous, non-overlapping frame ranges, ensuring every frame is accounted for and no frame overlaps between ranges. Each frame range should be no less than 13 frames.
2. Use changes in the person's actions or movements as the primary criterion for dividing frame ranges. For example, if the person starts walking in one frame and stops walking in another, these two events should belong to separate frame ranges. Avoid dividing frames arbitrarily by frame count.

Description Content:
1. Focus solely on the main person marked with ID "1".
2. Describe the person's appearance, actions, movements, and interactions with objects or other entities in the video.
3. Highlight any significant temporal changes in the person's actions, movements, or appearance between frame ranges.
4. Avoid describing background details, emotions, or the atmosphere.

Writing Style:
1. Be concise and accurate, with each description containing no more than 5 sentences.
2. Ensure each description is distinct and coherent, while maintaining temporal continuity across frame ranges.
3. Do not mention the numeric ID "1" or refer generically to "the main person." Instead, directly describe the person's actions.
4. Do not mention "the background", "the camera" or "the setting".

Output Format:
Write the output in the following format:
from [start_frame] to [end_frame]: [Description].
from [start_frame] to [end_frame]: [Description].
Ensure every frame in the video segment is assigned to a single frame range, and the frame ranges accurately reflect the person's actions.

**Example Output:**
When the max end_frame id is 57:
from 0 to 23: The person is walking steadily forward, holding a black briefcase in his right hand. His head turns slightly to the left, as if looking at something nearby.
from 24 to 41: The person stops walking and sets the briefcase down on the ground. He bends down and appears to adjust something on the briefcase.
from 42 to 57: The person straightens up, picks up the briefcase, and begins walking again, this time at a faster pace. His posture remains upright, and he glances briefly to the right.

The max end_frame id is *{frame_num-1}*. Please provide the frame range within this limit.

*Figure 12.* Instruction format for guiding Gemini2.5pro to generate detailed and distinct action descriptions, the *italicized* part are placeholders for the text inputs.

**Task:**

You are a helpful assistant designed to generate detailed and context-aware questions from video segment descriptions that include timestamps. For each input, generate the following three natural language questions:

1. A combined question that focuses on both location (where) and time (when). This question should not include timestamps. And this question should try to cover the details from the video description as many as possible, particularly the person's actions.

2. A where-only question, directly referencing the timestamp range in the format: "Where is [subject] during [start time] to [end time]?" Make sure to include details about the person's clothing, actions, or other relevant features that are described in the caption to make the question more specific and location-based.

3. A when-only question, focusing solely on the time aspect and not using the raw timestamp. Describe the person's action in detail and make sure the time-related question helps pinpoint the event or action in the video at a specific moment.

All questions must be specific and answerable based on the description. Avoid generic phrasing.

**Example Input:**

31.83s - 48.47s: The person wearing a black shirt is standing by the kitchen window early in the morning as sunlight streams in. She checks her phone and glances outside.

**Example Output:**

Combined question: Where is the woman in black when she checks her phone early in the morning by the kitchen window?

Where question: Where is the woman wearing a black shirt standing during 31.83s to 48.47s?

When question: When does the woman in black check her phone by the kitchen window?

---

**Input:**

*{Description of Shot}*

**Output:**

*Figure 13.* Instruction format for guiding LLaMA3.1 to generate three types of questions to construct LoomBench, the *italicized* part are placeholders for the text inputs.

