# OpenReview forum: "VideoLoom: A Video Large Language Model for Joint Spatial-Temporal Understanding"
_ICML.cc/2026/Conference — ICML 2026 regular_

### Official Review · Reviewer_E7vy · 2026-03-07

**Soundness:** 4
**Presentation:** 3
**Significance:** 2
**Originality:** 2
**Overall Recommendation:** 4
**Confidence:** 4

**Summary:**

This paper proposes (1) **VideoLoom**, a unified Video LLM for joint spatial-temporal understanding, together with (2) **LoomData-8.7k** and **LoomBench**. LoomData-8.7k is a character-centric dataset with temporally grounded action descriptions and mask-level spatial annotations, while LoomBench evaluates three settings — When, Where, and Combined — and introduces the J&Fbi-fore metric for the combined setting.

For the model, VideoLoom combines SlowFast-style visual tokenization with an MLLM-SAM2 architecture. Fast tokens provide broader temporal coverage, slow tokens preserve high-resolution spatial detail, the MLLM predicts timestamp-related outputs, and a special [SEG] token conditions SAM2 to generate spatial masklets. The training data is constructed through a multi-stage automatic pipeline including shot partition, spatial mask annotation/tracking, shot merging, and timestamp-aligned action description generation.

Experimentally, the paper reports strong results on both temporal and spatial benchmarks, including temporal grounding and referring video segmentation, and shows that VideoLoom outperforms a compositional TimeSuite+Sa2VA baseline on LoomBench Combined questions. Overall, the paper presents a coherent dataset-model-benchmark package for fine-grained spatial-temporal video understanding.

**Compliance With Llm Reviewing Policy:**

Affirmed.

**Final Justification:**

I can now better appreciate the paper as a benchmark/task-driven contribution, and the rebuttal and the further discussions meaningfully improved my assessment on that front. In particular, the clarification and reframing around the benchmark/task largely address my main concern there, which is why I updated my score. My remaining reservation is primarily on the method side: I still view the model contribution as somewhat incremental, and I do not think this aspect is fully resolved by the rebuttal. I therefore encourage the authors to position the paper more explicitly around its benchmark/dataset contribution and to state the method-side limitations more carefully.

**Key Questions For Authors:**

My overall impression is that the work is incremental but reasonably sound. The model can be viewed as a SlowFast-tokenized MLLM with a SAM2 / `[SEG]` grounding branch, while the dataset/benchmark largely combines spatial and temporal supervision into a unified setting. The experiments are comprehensive and the design is carefully justified. That said, despite being somewhat niche, this is still a solid contribution if the paper can better clarify the motivation, the added value of joint modeling, and the extent to which the method goes beyond the closest SlowFast-style baselines.

**Limitations:**

yes

**Strengths And Weaknesses:**

### Strengths
- The paper proposes a unified Video LLM for joint spatial-temporal understanding and shows strong results over a compositional temporal-then-spatial baseline on LoomBench Combined questions.
- The paper also introduces LoomData-8.7k and LoomBench, providing a new character-centric training set and testbed for fine-grained spatial-temporal video understanding.

### Weaknesses
- **For the model, the overall contribution still feels incremental relative to the closest SlowFast-style video-LLM literature.** The paper discusses closely related SlowFast-style approaches, but the core method still appears structurally close to a SlowFast-tokenized MLLM augmented with a SAM2 / `[SEG]` grounding branch. The current ablations suggest that the proposed components are useful in aggregate, but the paper still lacks a direct comparison against **SlowFast-LLaVA** or a sharper decomposition of how much of the gain comes from the shared SlowFast-style backbone idea versus the added grounding branch / training formulation.

- **For the benchmark/task framing, the main conceptual claim is still only partially convincing.** While the paper is framed around *joint* spatial-temporal understanding, LoomBench is still organized around **When / Where / Combined** questions, and the current empirical evidence mainly shows that the unified model outperforms a temporal-then-spatial pipeline baseline (Table 4). This is useful, but it does not yet explain **why** joint modeling is necessary or what specific complementary information is exchanged between temporal and spatial reasoning. I would find the contribution significantly more convincing with a more controlled analysis of (i) what spatial information improves temporal grounding and vice versa, and (ii) why the compositional temporal→spatial pipeline fails.

---

> ### Author Rebuttal · Authors · 2026-03-31
>
> **Q1: Direct comparison against SlowFast-LLaVA**
>
> Although both methods adopt a SlowFast-style visual token design, VideoLoom and SlowFast-LLaVA differ substantially in both motivation and implementation. 1) SlowFast-LLaVA treats SlowFast tokens as a training-free solution to balance the visual load, only using them as a token compression method. In contrast, VideoLoom introduces SlowFast tokens to effectively balance temporal coverage and spatial precision, meeting the diverse visual granularities required for spatial-temporal tasks. 2) VideoLoom is the first to extend SlowFast tokens to fine-grained spatial-temporal modeling by interleaving Fast tokens with timestamps. To the best of our knowledge, we are the first to demonstrate its effectiveness in unifying spatial-temporal tasks and improving joint training.
>
> **Q2: Sharper decomposition of the gains from the SlowFast-style backbone versus the added grounding branch**
>
> The added grounding branch integrates mask-level perception into the MLLM in an end-to-end manner, providing a structural foundation. However, this direct integration results in a performance trade-off, which motivates us to design the SlowFast-style method.
>
> We have conducted ablation studies in Table 5 of our paper. Below we cite part of it:
>
> |Setting|mIoU on Charades|J&F on ReVOS|
> |---|---|---|
> |Spatial (Slow)|-|59.5|
> |Temporal (Fast)|55.8|-|
> |Joint (Slow)|38.6|58.9|
> |Joint (Fast)|54.3|56.8|
> |Joint (Slow/Fast)|54.0|58.8|
> |Joint (SlowFast)|56.5|60.0|
>
> Comparing Temporal (Fast) and Joint (Fast), it is evident that introducing the SEG token leads to a decline in temporal performance. Compared to other joint designs, introducing slowfast tokens achieves the optimal results under joint training. Most notably, our Joint (SlowFast) not only eliminates the performance decline but also outperforms standalone spatial or temporal models.
>
> In summary, while the grounding branch establishes the architectural capability for spatial perception, the SlowFast visual tokens are the key to unifying spatial-temporal tasks and achieving joint understanding.
>
> **Q3: Why joint modeling is necessary and what specific complementary information is exchanged between temporal and spatial reasoning**
>
> Thanks for your valuable insights. To further illustrate the advantages of joint modeling, we provide visualization examples on https://anonymous.4open.science/r/icml_re-BC412/spatial_temporal_case.pdf and analyze from two critical perspectives:
>
> **(i) What spatial information improves temporal grounding and vice versa**
>
> **Spatial→Temporal:** Spatial perception aggregates high-level semantics onto the SEG token, enhancing target awareness. Since action grounding often involves object interactions, spatial information (e.g., object occurrence or relative positions) can improve the localization. In CASE A, joint modeling leverages the spatial clues (the spatial relationship between "the man" and "the mini-trampoline"), enabling VideoLoom to identify more precise action boundaries.
>
> **Temporal→Spatial:** Temporal grounding strengthens the capability to associate queries with video frames. This holistic frame-level identification provides coarse-to-fine guidance for region-level identification, facilitating precise segmentation. In CASE B, while both models locate similar intervals, only VideoLoom accurately identifies the correct "cowboy" in high-motion scenes, demonstrating superior semantic alignment.
>
> **(ii) Why the compositional pipeline fails**
>
> First, disjointed stages are strictly limited by individual model ceilings. Without joint modeling, they cannot exchange information to resolve ambiguities (as seen in Cases A & B). Second, the cascaded pipeline can compound these errors. As seen in CASE C, temporal offsets force the spatial model to process "empty" frames, yielding distorted SEG tokens and invalid masks.
>
> In summary, joint spatial-temporal modeling is therefore essential, facilitating the coherent spatial-temporal understanding with information exchange across both dimensions. And the design of Videoloom effectively leverages the complementarity to unlock spatial-temporal capabilities that fundamentally surpass the limitations of compositional pipelines.
>
> **Q4: Better clarify the motivation, the added value of joint modeling, and the extent to which the method goes beyond the closest SlowFast-style baselines**
>
> Our motivation is to unlock joint spatial-temporal understanding for video LLMs, since accurate perception in complex real-world scenarios often requires reasoning jointly over where and when. Existing methods, however, typically emphasize either spatial perception or temporal understanding in isolation, struggling to establish coherent spatial-temporal associations.
>
> We demonstrate the value of joint modeling in Q3. Regarding the closest SlowFast-style baselines, we clarify in Q1 that our method goes beyond them in both motivation and design. We will make these points clearer in the revised paper.

---

> > ### Author Rebuttal · Reviewer_E7vy · 2026-04-03
> >
> > Overall, the rebuttal makes the paper more convincing to me, especially on the benchmark/task-framing side. The added discussion largely addresses my main concern there and increases my confidence in the benchmark/dataset contribution. My remaining hesitation is primarily on the method side, where I still find the contribution somewhat incremental and the comparison to the closest SlowFast-style baseline incomplete. As a result, I now view the submission as a genuine borderline paper, closer to the bar than before but still with some uncertainty on the method side.

---

> > > ### Author Response · Authors · 2026-04-06
> > >
> > > Thank you for your careful and constructive review. We appreciate your recognition of the value of our task and benchmark.
> > >
> > > While VideoLoom adopts a SlowFast-style visual token design, its motivation and implementation are fundamentally different. In previous video-LLM works, SlowFast is primarily a tool for token compression[1, 2] or general temporal representation[3], focusing on efficiency. In contrast, we adopt SlowFast as a dedicated mechanism for joint spatial-temporal modeling, explicitly addressing the inherent conflicts between spatial perception and temporal understanding. In terms of implementation, VideoLoom is the first to interleave Fast tokens with timestamps, achieving a unified training paradigm for joint spatial-temporal tasks. This is not a direct extension of previous baselines, but rather a task-oriented, principled solution that facilitates information exchange between spatial and temporal dimensions, effectively avoiding the performance degradation typical of naive joint training.
> > >
> > > We clarify that our contribution lies not in introducing a completely new model architecture, but in formulating a joint spatial-temporal understanding task and presenting a unified, practical solution encompassing data, model, and benchmark. Both aspects are non-trivial: the proposed task identifies key limitations of prior work and extends the problem along multiple dimensions (e.g., granularity[4, 5] and task integration[6, 7]), while our solution provides a coherent framework that effectively addresses two key bottlenecks (i.e., the lack of large-scale joint spatial-temporal annotations, and the absence of unified models and benchmarks). We believe this co-design of a new problem formulation and its holistic solution represents a systematic advancement that cannot be achieved by simple incremental extensions of prior work, which underscores that our contribution goes beyond isolated architectural modifications and instead establishes a foundation that significantly advances video understanding in complex real-world scenarios.
> > >
> > > We appreciate your time and thoughtful feedback, and hope our clarifications could contribute positively to your final evaluation.
> > >
> > > [1] Xu, Mingze, et al. "Slowfast-llava: A strong training-free baseline for video large language models." arXiv preprint arXiv:2407.15841.
> > >
> > > [2] Xu, Mingze, et al. "Slowfast-llava-1.5: A family of token-efficient video large language models for long-form video understanding." arXiv preprint arXiv:2503.18943.
> > >
> > > [3] Huang, De-An, et al. "Lita: Language instructed temporal-localization assistant." ECCV 2024.
> > >
> > > [4] Li, Hongyu, et al. "Llava-st: A multimodal large language model for fine-grained spatial-temporal understanding." CVPR 2025.
> > >
> > > [5] Deng, Andong, et al. "Motion-grounded video reasoning: Understanding and perceiving motion at pixel level." CVPR 2025.
> > >
> > > [6] Zeng, Xiangyu, et al. "Timesuite: Improving mllms for long video understanding via grounded tuning." ICLR 2025.
> > >
> > > [7] Yan, Cilin, et al. "Visa: Reasoning video object segmentation via large language models." ECCV 2024.
> > >
> > > **Looking forward to your discussion:** Dear reviewer, thank you again for your insightful comments on our paper, and we genuinely hope that our clarifications could address your concerns. As the discussion is about to end, we are sincerely looking forward to your feedback. Please feel free to contact us if you have any further inquiries.

---

### Official Review · Reviewer_4KZD · 2026-03-08

**Soundness:** 3
**Presentation:** 2
**Significance:** 3
**Originality:** 2
**Overall Recommendation:** 4
**Confidence:** 2

**Summary:**

This paper proposes a model that can handle both fine-grained temporal understanding and pixel-level spatial perception within a unified framework, addressing a gap in joint spatio-temporal reasoning. The authors design “fast tokens” and “slow tokens,” successfully balancing the need for temporally dense tokens and spatially high-resolution tokens under limited computational resources. Overall, the design motivation is reasonable.

However, SlowFast-style strategies and SAM-based spatial modeling have already received considerable attention in recent VLM research [1,2], and the additional 8.7K data contribution does not seem sufficient to substantially elevate the impact of this work. Therefore, I reckon the overall contribution and novelty of the paper are not yet strong enough. I will finalize my score after discussion with the authors to ensure this score is accurate.

**References:**

[1] Fu, Shenghao, et al. "Love-r1: Advancing long video understanding with an adaptive zoom-in mechanism via multi-step reasoning." arXiv preprint arXiv:2509.24786 (2025).
[2] Yuan, Haobo, et al. "Sa2va: Marrying sam2 with llava for dense grounded understanding of images and videos." arXiv preprint arXiv:2501.04001 (2025).

**Compliance With Llm Reviewing Policy:**

Affirmed.

**Final Justification:**

I appreciate the authors for solving my concerns, I will increase my score.

Although the authors emphasize that it is a pioneering effort in the spatial-temporal domain, the overall contribution and novelty are still limited as they have already been widely explored, I wish the authors can make better clarification in the future version.

**Key Questions For Authors:**

1. Given that the generation of LoomData heavily relies on detecting a single main character, how does VideoLoom perform when handling queries involving interactions among multiple entities (e.g., “the player in blue who is passing the ball to the person in red”)? Although zero-shot evaluation was conducted on RefDavis17, could you provide additional qualitative analysis on scenarios involving multi-object occlusion or non-human primary entities, such as animals or vehicles?

2. VideoLoom adopts a decoupled architecture. If, in a complex occlusion scenario, SAM2 fails to generate a valid mask based on the [SEG] token, the system does not appear to include a feedback mechanism to inform the MLLM that “no valid mask was produced,” so that it can revise its initial temporal prediction. Have the authors considered addressing this limitation?

**Limitations:**

Please see above.

**Strengths And Weaknesses:**

**Strengths:**
1. VideoLoom proposes a model capable of handling both fine-grained temporal understanding and pixel-level spatial perception within a unified framework, addressing a gap in joint spatio-temporal reasoning.

2. The authors design “fast tokens” and “slow tokens,” successfully balancing the need for temporally dense tokens and spatially high-resolution tokens under limited computational resources.

3. The paper proposes a Bidirectional Foreground J&F metric that excludes interference from irrelevant background regions, and the motivation behind it is reasonable.

**Weaknesses:**
1. The annotation pipeline of LoomData-8.7K mainly relies on GroundingDINO to detect the highest-scoring bounding box as the main character. This design leads to training data that is overly concentrated on single-person or single-object scenarios. In real-world complex videos, there are often multiple entities interacting in intricate ways. As a result, the model’s generalization ability to multi-agent scenarios or non-human primary targets may be limited, and the scope of problems covered by the dataset may not be sufficiently broad.

2. Given the limited generalization capacity of the dataset, although the paper proposes a unified spatio-temporal understanding architecture, both the SlowFast strategy and the use of SAM with special tokens are relatively common techniques. As such, the architectural contribution appears incremental, which further weakens the overall novelty and impact of the work.

---

> ### Author Rebuttal · Authors · 2026-03-31
>
> **Q1: The model’s generalization ability to multi-agent scenarios or non-human primary targets may be limited, and the scope of problems covered by the dataset may not be sufficiently broad.**
>
> We provide ablation results on LoomData in Q3 and Sec. D.2 of the supplementary material, which demonstrates the generalization capability of both VideoLoom and LoomData beyond the core annotation setting.
>
> We agree that the current scope of LoomData can be further broadened. With more advanced tracking models such as SAM3[1], we plan to extend our annotation pipeline to non-human objects and multi-instance scenarios in future work. We also plan to enrich the annotation types, for example, by introducing HOI annotations, so that the dataset can better cover a wider range of video understanding scenarios.
>
> [1] Carion, Nicolas, et al. "Sam 3: Segment anything with concepts." ICLR 2026.
>
> **Q2: The architectural contribution appears incremental, which further weakens the overall novelty and impact of the work.**
>
> We would like to clarify that the main novelty of our work does not lie in introducing a completely new architectural primitive, but in providing a unified and practical solution for joint spatial-temporal understanding, a setting that remains underexplored despite its importance in real-world video understanding. In complex video scenarios, accurate understanding often requires reasoning jointly over where and when. However, existing approaches typically focus on either the spatial or the temporal dimension in isolation, struggling to establish coherent spatial-temporal associations.
>
> To address this gap, we contribute a complete framework spanning data, model, and benchmark: LoomData provides joint spatial-temporal supervision, VideoLoom enables joint modeling, and LoomBench evaluates this capability in a targeted manner. We agree that components such as SAM with special tokens and the SlowFast strategy have individually appeared in prior work, but we are the first to integrate them into a single framework for joint spatial-temporal modeling and demonstrate the effectiveness in this setting.
>
> **Q3: How does VideoLoom perform when handling queries involving interactions among multiple entities?**
>
> Although LoomData focuses on a single main character, its annotations are not limited to single-entity descriptions. Many samples explicitly involve interactions between multiple entities, for example: "The man lunges forward to engage with a woman, grappling with her. He quickly spins her around before bending at the waist." This means LoomData still provides supervision for relational understanding in multi-entity scenarios.
>
> To quantitatively assess this capability, we further select 275 interaction-centric queries from MeVIS, such as "rabbit leaping over another rabbit", and conduct an ablation study on LoomData.
>
> |Method|J|F|J&F|
> |---|---|---|---|
> |w/o LoomData|59.1|68.2|63.6|
> |Ours|60.9|69.5|65.2|
>
> The results show that incorporating LoomData brings a notable improvement of +1.6 J&F on these interaction queries. It demonstrates that LoomData provides useful supervision not only for single-entity localization but also for relational reasoning in multi-entity interaction scenarios.
>
> **Q4: Additional qualitative analysis on scenarios involving multi-object occlusion or non-human primary entities.**
>
> We provide additional qualitative visualizations at https://anonymous.4open.science/r/icml_re-BC412/complex_seg_case.pdf. These examples are all drawn from ReVOS and cover both referring and reasoning VOS, including challenging scenarios such as multi-object interactions, multi-object occlusion, and non-human targets.
>
> As shown in Case A and Case B, VideoLoom can accurately localize targets in complex interaction scenes and correctly identify distinctive non-human objects such as vehicles. In Case C, VideoLoom also shows competitive performance in tracking occluded targets under severe multi-object occlusion.
>
> These qualitative results, together with our ablation studies on complex spatial-temporal scenarios, further support the effectiveness and generalization ability of VideoLoom in real-world settings.
>
> **Q5: Discussion on the feedback mechanism.**
>
> Thank you for the suggestion. We agree that the current system does not explicitly provide feedback to the MLLM when SAM2 fails to generate a valid mask. A promising future direction is to introduce a reflection mechanism after mask generation. The model could first evaluate the validity of the predicted mask, e.g., using rule-based criteria, an external instance-understanding model, or a rejection token, and then recursively revise its temporal localization based on this feedback. If the mask is only partially valid, the interval can be refined; if it is invalid throughout, the segment can be excluded, and grounding can be re-run.
>
> We believe this feedback-driven design could make VideoLoom more robust in challenging cases, and we will clarify this in the revised paper.

---

> > ### Author Rebuttal · Reviewer_4KZD · 2026-04-02
> >
> > I appreciate the authors for solving my concerns, I will increase my score.
> >
> > Although the authors emphasize that it is a pioneering effort in the spatial-temporal domain, the overall contribution and novelty are still limited as they have already been widely explored, I wish the authors can make better clarification in the future version.

---

> > > ### Author Response · Authors · 2026-04-02
> > >
> > > Thanks a lot for your quick feedback and considering to increase your score!
> > >
> > > Understanding complex spatial-temporal events in videos is highly significant yet underexplored, as most existing approaches focus on either spatial or temporal in isolation. To address this gap, we introduce the task of joint spatial-temporal video understanding, which goes beyond prior isolated formulations by requiring models to reason jointly over where and when. We identify two key bottlenecks in this direction: the lack of large-scale joint spatial-temporal annotations, and the absence of unified architectures designed specifically for joint training and evaluation. To overcome these challenges, we present the first comprehensive suite comprising data, model, and benchmark. This enables a principled and unified approach to joint spatial–temporal modelling and establishes a strong baseline that fundamentally surpasses the limitations of model-in-series pipelines.
> > >
> > > We hope VideoLoom could serve as a foundation for advancing spatial-temporal perception and reasoning in video LLMs, and inspire further architectural innovations in this promising area.
> > >
> > > We will incorporate clearer explanations of these aspects in the revised version.
> > >
> > > **Looking forward to your discussion:** Dear reviewer, thank you again for your insightful comments on our paper, and we genuinely hope that our clarifications could address your concerns. As the discussion is about to end, we are sincerely looking forward to your feedback. Please feel free to contact us if you have any further inquiries.

---

### Official Review · Reviewer_Yp7H · 2026-03-16

**Soundness:** 3
**Presentation:** 3
**Significance:** 3
**Originality:** 3
**Overall Recommendation:** 4
**Confidence:** 5

**Summary:**

This paper presents VideoLoom, a video LLM designed to jointly understand spatial and temporal dimensions when processing complex video events. To achieve this, the authors propose a three-part solution: 1) LoomData-8.7k Dataset: A large-scale, character-centric dataset featuring temporally aligned action descriptions and fine-grained spatial masks with mask-level annotations; 2) VideoLoom Architecture: The model integrates MLLM with SAM2 and introduces SlowFast Visual Tokens; 3) LoomBench Benchmark: A new evaluation suite containing more than 1,400 queries. It also introduces a new metric for comprehensive assessment of temporal grounding, spatial segmentation, and their combined performance. Experimental results show that VideoLoom achieves sota performance across diverse visual benchmarks.

**Compliance With Llm Reviewing Policy:**

Affirmed.

**Key Questions For Authors:**

Please see the weaknesses.

**Limitations:**

yes

**Strengths And Weaknesses:**

### Strengths

- VideoLoom is the first to achieve fine-grained temporal understanding and mask-level spatial perception within a unified framework.

- This design effectively balances computational cost and input granularity. By spatially downsampling dense frames into fast tokens while preserving high resolution for keyframes as slow tokens, it resolves the mismatch between the resolution and frame-sampling requirements of spatial-temporal tasks.

- LoomData fills the gap in datasets that simultaneously provide high-quality temporal and spatial annotations. The “Combined” category in LoomBench better reflects real-world scenarios.

- The model performs strongly across several mainstream tasks. In addition, the 4B version maintains high performance while delivering fast inference speed and low memory usage.

### Weaknesses

- Visualization analysis shows that in highly complex joint understanding scenarios, such as identifying the n-th occurrence of a specific action or reasoning over very long action sequences, the model may occasionally produce inaccurate spatial-temporal localization.

- Although experiments demonstrate its generalization ability, LoomData is currently centered mainly on the “Person” category, leaving room to further expand fine-grained interaction annotations for non-human objects.

- The performance of the baseline model InternVL3 is missing. For example, in Table 1, its performance could be readily evaluated by directly prompting it to output the temporal ranges corresponding to the query.

- More base models should be included to better assess the quality of the proposed data, such as Qwen series. In addition, I suggest that the authors evaluate VideoLoom on TimeLens-Bench [1], as it improves the annotation quality of the original benchmarks.

[1] TimeLens: Rethinking Video Temporal Grounding with Multimodal LLMs. CVPR 2026.

---

> ### Author Rebuttal · Authors · 2026-03-31
>
> **Q1: Model may occasionally produce inaccurate spatial-temporal localization in highly complex joint understanding scenarios.**
>
> Thank you for pointing this out. We have already discussed representative failure cases in highly complex joint understanding scenarios in Sec. E.2 of the supplementary material. These cases suggest that the current model can still struggle with accurate spatial-temporal localization when the query requires more complex reasoning across both spatial and temporal dimensions.
>
> A promising direction to address this issue is to strengthen the reasoning process. In particular, we plan to introduce Chain-of-Thought (CoT) reasoning for temporal grounding, following Time-R1[1], and further improve reasoning ability through reinforcement learning. We expect this to improve performance on the challenging failure cases identified above. We will make this discussion clearer in the revised paper.
>
> [1] Wang, Ye, et al. "Time-r1: Post-training large vision language model for temporal video grounding." NeurIPS 25.
>
> **Q2: LoomData is currently centered mainly on the “Person” category, leaving room to further expand fine-grained interaction annotations for non-human objects.**
>
> Thank you for pointing this out. When constructing LoomData, we focus mainly on the human category as they carry rich action and semantic information[1, 2]. In addition, for an automatic annotation pipeline, humans usually have clearer identity and tracking cues, which makes it easier to build accurate and temporally consistent spatial-temporal annotations throughout the video.
>
> We agree that extending LoomData beyond the human category is an important direction. With the emergence of more advanced tracking models such as SAM3[3], we plan to extend our annotation pipeline to non-human objects in the future, thereby broadening the coverage and generality of LoomData.
>
> [1] Gu, Chunhui, et al. "Ava: A video dataset of spatio-temporally localized atomic visual actions." CVPR 2018.
>
> [2] Tang, Zongheng, et al. "Human-centric spatio-temporal video grounding with visual transformers." TCSVT 2021.
>
> [3] Carion, Nicolas, et al. "Sam 3: Segment anything with concepts." ICLR 2026.
>
> **Q3: The performance of the baseline model InternVL3.**
>
> Thank you for your suggestions. Below, we compare VideoLoom with InternVL3, which demonstrates that our method outperforms the backbone model we built upon by a clear margin. We will include these results in the revised paper.
>
> | Method | R1@0.5 on Charades | R1@0.7 on Charades | SODA_c on YC2 | CIDEr on YC2 | F1 on YC2 | mAP on QVHL | HIT@1 on QVHL |
> | --- | --- | --- | --- | --- | --- | --- | --- |
> | InternVL3 | 24.8 | 12.3 | 0.3 | 0.8 | 3.7 | 13.6 | 17.0 |
> | VideoLoom | 70.0 | 48.3 | 7.3 | 41.5 | 33.6 | 27.5 | 63.3 |
>
> **Q4: More base models should be included to better assess the quality of the proposed data, such as Qwen series.**
>
> Thank you for your suggestion. To better assess the quality of LoomData, we conduct ablation experiments on the Qwen series, specifically using Qwen3VL-4B as the backbone. For simplicity, we omit the VQA data and, consistent with Table 6, use joint training on existing temporal and spatial datasets as the baseline.
>
> | Dataset | mIoU on Charades | J&F on ReVOS | tIoU on LoomBench | bi-fore J&F on LoomBench |
> | --- | --- | --- | --- | --- |
> | Baseline | 43.0 | 62.2 | 21.7 | 27.6 |
> | +LoomData | 44.5 | 62.5 | 26.7 | 35.1 |
>
> As shown, after incorporating LoomData, the Qwen model achieves stable gains on all benchmarks, with a +7.5 bi-fore J&F improvement on LoomBench. This further demonstrates the effectiveness of LoomData in enhancing spatial–temporal understanding.
>
> **Q5: Evaluation on TimeLens-Bench.**
>
> Thanks for your suggestion. We evaluate the performance of VideoLoom on TimeLens-Bench and compare it with advanced temporal models.
>
> | Method | mIoU on Charades | mIoU on Anet | mIoU on QVHighlights | Avg. |
> | --- | --- | --- | --- | --- |
> | TimeSuite | 38.1 | 19.8 | 21.7 | 24.7 |
> | Grounded-VideoLLM | 30.0 | 30.0 | 33.4 | 30.6 |
> | TRACE | 27.1 | 32.7 | 39.0 | 32.0 |
> | Time-R1 | 36.6 | 33.1 | 49.2 | 39.5 |
> | VideoLoom | 43.1 | 31.8 | 44.4 | 39.8 |
>
> The evaluation result shows that VideoLoom achieves an average mIoU of 39.8, outperforming both unified temporal models (e.g., TimeSuite and TRACE) and TVG task-specific models (e.g., Grounded-VideoLLM and Time-R1).
>
> We also include the evaluation results on TimeLens-Bench in the ablation experiments to ensure a more accurate assessment of the temporal performance gains achieved by LoomData.
>
> | Dataset | mIoU on Charades | mIoU on Anet | mIoU on QVHighlights | Avg. |
> | --- | --- | --- | --- | --- |
> | Baseline | 35.4 | 21.3 | 36.7 | 31.1 |
> | +VQA | 35.8 | 22.4 | 37.7 | 32.0 |
> | +LoomData | 39.3 | 28.9 | 43.6 | 37.3 |
>
> Training with LoomData significantly improves temporal grounding. These results further demonstrate that LoomData could provide high-quality supervision for temporal comprehension.

---

> > ### Author Rebuttal · Reviewer_Yp7H · 2026-04-04
> >
> > Thank you for the response. My concern has been fully addressed. I suggest that the authors incorporate these discussions into the final version.

---

> > > ### Author Response · Authors · 2026-04-06
> > >
> > > Thank you very much for your time. We truly appreciate your constructive feedback and are pleased to hear that your concerns have been addressed. We will incorporate these discussions and evaluations in the revised version.

---

### Official Review · Reviewer_U3Pp · 2026-03-16

**Soundness:** 3
**Presentation:** 3
**Significance:** 3
**Originality:** 2
**Overall Recommendation:** 4
**Confidence:** 4

**Summary:**

This paper addresses the limitations of existing video LLMs, which typically focus on either temporal or spatial understanding separately. To tackle this problem, the authors propose VideoLoom, a unified framework that integrates temporal localization and spatial segmentation within a single model. Specifically, VideoLoom combines a multimodal LLM with a segmentation model via a special [SEG] token, and the SlowFast visual token design balances dense temporal coverage with high-resolution spatial detail. Besides, the authors also construct the instruction tuning dataset LoomData-8.7k, which includes aligned temporal descriptions and spatial mask tracklets generated through an automatic annotation pipeline. Experimental results across multiple temporal grounding, video segmentation, image segmentation, and unified benchmarks demonstrate that VideoLoom achieves state-of-the-art or competitive performance, outperforming specialized methods on several tasks and showing strong gains on the proposed LoomBench.

**Compliance With Llm Reviewing Policy:**

Affirmed.

**Final Justification:**

I still think the novelty of this paper is limited sice previous research has explored this topic deeply. But this work is solid in experiment, thus I still recommend to accept it.

**Key Questions For Authors:**

Please check the weakness.

**Limitations:**

Yes

**Strengths And Weaknesses:**

Strengths:
1. The paper is generally clear and logically structured.
2. The proposed benchmarks and datasets may stimulate future research toward integrated video understanding/reasoning systems.
3. The proposed new metric can jointly evaluate the model from both spatial and temporal perspective.
4. Extensive experiments demonstrate the effectiveness of the proposed method.

Weakness:
1. Similar task is proposed in previous work, e.g., VideoLISA[1], GroundMoRe[2], etc.
2. It looks like training with LoomData-8.7k improves the performance on the LoomBench a lot, which has limited performance gain on other benchmarks. This might potentially limit the usage of this training dataset in general video segmentation scenarios. And the reason behind also need more discussion (e.g., why with zero performance gain on YTVOS, which is also a video seg benchmark.)
3. In line 379, the "joint (Slow/Fast) setting" might need further clarification. What is the difference between this one and "joint (SlowFast) setting".

[1] Bai et al., One Token to Seg Them All: Language Instructed Reasoning Segmentation in Videos
[2] Deng et al., Motion-grounded Video Reasoning: Understanding and Perceiving Motion at Pixel Level

---

> ### Author Rebuttal · Authors · 2026-03-31
>
> **Q1: Similar task is proposed in previous work, e.g., VideoLISA, GroundMoRe, etc.**
>
> The proposed task, joint spatial-temporal video understanding, differs fundamentally from previous work, such as VideoLISA and GroundMoRe. We aim to evaluate the joint modeling of fine-grained spatial-temporal tasks and emphasize timestamped, mask-level localization of queries across long-form videos. In our proposed task, the model must handle joint spatial-temporal queries that integrate "when" and "where" to determine the precise time intervals and spatial mask sequences where the target is located. Furthermore, the query target appears on average for only 20.9% of the entire video, and the queries involve complex interactions and temporal relationships. This requires the model to jointly model space and time, which goes beyond the scope of existing tasks.
>
> In contrast, VideoLISA and GroundMoRe clearly focus on spatial perception. Neither involves the modeling of precise timestamps, and both handle only fixed-length frame sequences and videos of 5–15 seconds, limiting the evaluation of temporal grounding and reasoning segmentation capabilities in complex spatial-temporal scenarios.
>
> **Q2: It looks like training with LoomData-8.7k improves the performance on the LoomBench a lot, which has limited performance gain on other benchmarks, e.g., RefYTVOS.**
>
> Thank you for the question. The baseline model (the 1st row of Table 6) is trained on mixed data that already covers the training sets of all evaluated benchmarks, including RefYTVOS, and therefore starts from a strong performance level. We believe the relatively limited gain on RefYTVOS is mainly due to the nature of the benchmark: its queries focus more on direct referring than on complex reasoning. Since the baseline has already been exposed to substantial referring-oriented supervision, adding a relatively small amount of LoomData naturally brings only limited additional benefit.
>
> In contrast, the greater improvement on ReVOS supports the effectiveness of LoomData in reasoning-intensive video segmentation scenarios. As an additional check, we increase the repetition of LoomData from 4 to 8, which still improves RefYTVOS from 70.3 to 70.5. This suggests that LoomData remains beneficial on RefYTVOS, although its effect is more pronounced on benchmarks with higher reasoning demands.
>
> As for the significant improvement on the proposed LoomBench, we believe this mainly stems from the unique role of LoomData. The baseline is trained jointly on datasets that provide either temporal-only or spatial-only annotations, but inconsistencies across annotation formats and data distributions can lead to suboptimal learning. LoomData complements this setup by providing joint spatial-temporal annotations on videos, enabling the model to learn coherent spatial-temporal associations. This capability is particularly important for complex scenarios requiring joint spatial-temporal localization, and therefore leads to substantial gains on LoomBench.
>
> **Q3: Further clarification for the "joint (Slow/Fast) setting"**
>
> Thanks for your advice! The specific implementation of the "joint (Slow/Fast) setting" involves inputting only fast visual tokens interleaved with timestamps for temporal tasks, and only slow visual tokens for spatial tasks, thereby enabling joint training of temporal and spatial tasks using different visual token formats. This setting can be viewed as a direct, mechanical combination of general training schemes for temporal and spatial tasks. In contrast, the "joint (SlowFast) setting" involves inputting both fast visual tokens interleaved with timestamps and slow visual tokens for both temporal and spatial tasks, thereby enabling joint training using the same visual token format. VideoLoom adopts the "joint (SlowFast) setting", effectively balancing temporal coverage and spatial precision. We will add this clarification in the revised paper.

---

> > ### Author Rebuttal · Reviewer_U3Pp · 2026-04-03
> >
> > Thanks for the explanation. However, I will maintain my score because the idea has already been explored by previous research. But I will still tend to accept this one because of the framework design (slowfast token) and the effectiveness of the training data.

---

> > > ### Author Response · Authors · 2026-04-06
> > >
> > > Thank you very much for your positive recommendation. We sincerely appreciate your recognition of the effectiveness of our framework design and the data we constructed.
> > >
> > > The proposed joint spatial-temporal task goes beyond the scope of previous research, and our comprehensive solution provides an effective approach to this challenging problem, advancing spatial-temporal perception and reasoning in video LLMs. We hope VideoLoom will stimulate further research in this promising direction.

---

### Decision · Program_Chairs · 2026-04-30

**Decision:**

Accept (regular)

**Comment:**

The paper proposes VideoLoom, a unified suite for joint spatial-temporal video understanding, which includes a character-centric dataset (LoomData), and a comprehensive evaluation benchmark (LoomBench). The submission received generally positive feedback, with reviewers noting its clarity and the effectiveness of the unified framework. Initial concerns were raised regarding the incremental nature of the architecture, the model's generalization capability, and the need for stronger justification for joint modeling over compositional pipelines. The authors provided a comprehensive rebuttal with additional experiments, and conduct a lot of analysis. Following the rebuttal, one reviewer increased their score, and most reviewers indicated that their primary concerns were addressed. While some reviewers still consider the methodological contribution somewhat incremental, they acknowledge that the paper provides a significant and practical contribution by formulating the joint spatial-temporal task and offering a holistic solution. Therefore, the AC recommends acceptance of this submission.